# Norm of Word Embedding Encodes Information Gain

**Momose Oyama**[1,3] **Sho Yokoi**[2,3] **Hidetoshi Shimodaira**[1,3]

[1]Kyoto University [2]Tohoku University [3]RIKEN AIP

oyama.momose@sys.i.kyoto-u.ac.jp,
yokoi@tohoku.ac.jp, shimo@i.kyoto-u.ac.jp

## Abstract

Distributed representations of words encode lexical semantic information, but what type of information is encoded and how? Focusing on the skip-gram with negative-sampling method, we found that the squared norm of static word embedding encodes the information gain conveyed by the word; the information gain is defined by the Kullback-Leibler divergence of the co-occurrence distribution of the word to the unigram distribution. Our findings are explained by the theoretical framework of the exponential family of probability distributions and confirmed through precise experiments that remove spurious correlations arising from word frequency. This theory also extends to contextualized word embeddings in language models or any neural networks with the softmax output layer. We also demonstrate that both the KL divergence and the squared norm of embedding provide a useful metric of the informativeness of a word in tasks such as keyword extraction, proper-noun discrimination, and hypernym discrimination.

## 1 Introduction

The strong connection between natural language processing and deep learning began with word embeddings (Mikolov et al., 2013; Pennington et al., 2014; Bojanowski et al., 2017; Schnabel et al., 2015). Even in today's complex models, each word is initially converted into a vector in the first layer. One of the particularly interesting empirical findings about word embeddings is that the norm represents the relative importance of the word while the direction represents the meaning of the word (Schakel and Wilson, 2015; Khodak et al., 2018; Arefyev et al., 2018; Pagliardini et al., 2018; Yokoi et al., 2020).

This study focuses on the word embeddings obtained by the skip-gram with negative sampling (SGNS) model (Mikolov et al., 2013). We show theoretically and experimentally that the Euclidean

| Top 10 | | Bottom 10 | |
|---|---|---|---|
| word | $\mathrm{KL}(w)$ | word | $\mathrm{KL}(w)$ |
| rajonas | 11.31 | the | 0.04 |
| rajons | 10.82 | in | 0.04 |
| dicrostonyx | 10.31 | and | 0.04 |
| dasyprocta | 10.27 | of | 0.05 |
| stenella | 10.24 | a | 0.07 |
| pesce | 10.22 | to | 0.09 |
| audita | 10.09 | by | 0.09 |
| landesverband | 10.05 | with | 0.10 |
| auditum | 9.96 | for | 0.10 |
| factum | 9.84 | s | 0.10 |

Table 1: Top 10 words and bottom 10 words sorted by the value of $\mathrm{KL}(w)$ in the text8 corpus with word frequency $n_w \geq 10$.

norm of embedding for word $w$, denoted as $\|u_w\|$, is closely related to the Kullback-Leibler (KL) divergence of the co-occurrence distribution $p(\cdot|w)$ of a word $w$ for a fixed-width window to the unigram distribution $p(\cdot)$ of the corpus, denoted as

$$\mathrm{KL}(w) := \mathrm{KL}(p(\cdot|w) \| p(\cdot)).$$

In Bayesian inference, the expected KL divergence is called information gain. In this context, the prior distribution is $p(\cdot)$, and the posterior distribution is $p(\cdot|w)$. The information gain represents how much information we obtain about the context word distribution when observing $w$. Table 1 shows that the 10 highest values of $\mathrm{KL}(w)$ are given by context-specific informative words, while the 10 lowest values are given by context-independent words.

Fig. 1 shows that $\|u_w\|^2$ is almost linearly related to $\mathrm{KL}(w)$; this relationship holds also for a larger corpus of Wikipedia dump as shown in Appendix G. We prove in Section 4 that the square of the norm of the word embedding with a whitening-like transformation approximates the KL divergence[1]. The main results are explained

---

[1]Readers who are interested in information-theoretic mea-

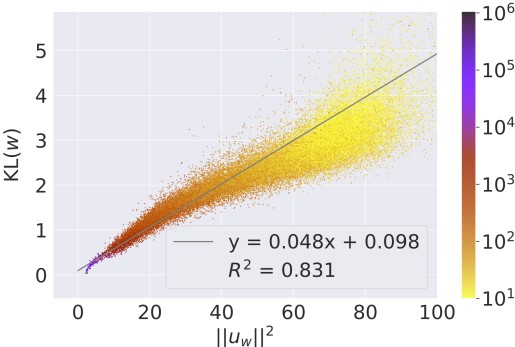

Figure 1: Linear relationship between the KL divergence and the squared norm of word embedding for the text8 corpus computed with 100 epochs. The color represents word frequency $n_w$. Plotted for all vocabulary words, but those with $n_w < 10$ were discarded. A regression line was fitted to words with $n_w > 10^3$. Other settings are explained in Section 4.2 and Appendix A.

by the theory of the exponential family of distributions (Barndorff-Nielsen, 2014; Efron, 1978, 2022; Amari, 1982).

Empirically, the KL divergence, and thus the norm of word embedding, are helpful for some NLP tasks. In other words, the notion of information gain, which is defined in terms of statistics and information theory, can be used directly as a metric of informativeness in language. We show this through experiments on the tasks of keyword extraction, proper-noun discrimination, and hypernym discrimination in Section 7.

In addition, we perform controlled experiments that correct for word frequency bias to strengthen the claim. The KL divergence is heavily influenced by the word frequency $n_w$, the number of times that word $w$ appears in the corpus. Since the corpus size is finite, although often very large, the KL divergence calculated from the co-occurrence matrix of the corpus is influenced by the quantization error and the sampling error, especially for low-frequency words. The same is also true for the norm of word embedding. This results in bias due to word frequency, and a spurious relationship is observed between word frequency and other quantities. Therefore, in the experiments, we correct the word frequency bias of the KL divergence and the norm of word embedding.

The contributions of this paper are as follows:

---

sures other than KL divergence are referred to Appendix B. The KL divergence is more strongly related to the norm of word embedding than the Shannon entropy of the co-occurrence distribution (Fig. 7) and the self-information $-\log p(w)$ (Fig. 8).

- We showed theoretically and empirically that the squared norm of word embedding obtained by the SGNS model approximates the information gain of a word defined by the KL divergence. Furthermore, we have extended this theory to encompass contextualized embeddings in language models.

- We empirically showed that the bias-corrected KL divergence and the norm of word embedding are similarly good as a metric of word informativeness.

After providing related work (Section 2) and theoretical background (Section 3), we prove the theoretical main results in Section 4. In Section 5, we extend this theory to contextualized embeddings. We then explain the word frequency bias (Section 6) and evaluate $KL(w)$ and $\|u_w\|^2$ as a metric of word informativeness in the experiments of Section 7.

## 2 Related work

### 2.1 Norm of word embedding

Several studies empirically suggest that the norm of word embedding encodes the word informativeness. According to the additive compositionality of word vectors (Mitchell and Lapata, 2010), the norm of word embedding is considered to represent the importance of the word in a sentence because longer vectors have a larger influence on the vector sum. Moreover, it has been shown in Yokoi et al. (2020) that good performance of word mover's distance is achieved in semantic textual similarity (STS) task when the word weights are set to the norm of word embedding, while the transport costs are set to the cosine similarity. Schakel and Wilson (2015) claimed that the norm of word embedding and the word frequency represent word significance and showed experimentally that proper nouns have embeddings with larger norms than function words. Also, it has been experimentally shown that the norm of word embedding is smaller for less informative tokens (Arefyev et al., 2018; Kobayashi et al., 2020).

### 2.2 Metrics of word informativeness

**Keyword extraction.** Keywords are expected to have relatively large amounts of information. Keyword extraction algorithms often use a metric of the "importance of words in a document" calculated by some methods, such as TF-IDF or word

co-occurrence (Wartena et al., 2010). Matsuo and Ishizuka (2004) showed that the $\chi^2$ statistics computed from the word co-occurrence are useful for keyword extraction. The $\chi^2$ statistic is closely related to the KL divergence (Agresti, 2013) since $\chi^2$ statistic approximates the likelihood-ratio chi-squared statistic $G^2 = 2n_w \text{KL}(w)$ when each document is treated as a corpus.

**Hypernym discrimination.** The identification of hypernyms (superordinate words) and hyponyms (subordinate words) in word pairs, e.g., *cat* and *munchkin*, has been actively studied. Recent unsupervised hypernym discrimination methods are based on the idea that hyponyms are more informative than hypernyms and make discriminations by comparing a metric of the informativeness of words. Several metrics have been proposed, including the KL divergence of the co-occurrence distribution to the unigram distribution (Herbelot and Ganesalingam, 2013), the Shannon entropy (Shwartz et al., 2017), and the median entropy of context words (Santus et al., 2014).

**Word frequency bias.** Word frequency is a strong baseline metric for unsupervised hypernym discrimination. Discriminations based on several unsupervised methods with good task performance are highly correlated with those based simply on word frequency (Bott et al., 2021). KL divergence achieved 80% precision but did not outperform the word frequency (Herbelot and Ganesalingam, 2013). WeedsPrec (Weeds et al., 2004) and SLQS Row (Shwartz et al., 2017) correlate strongly with frequency-based predictions, calling for the need to examine the frequency bias in these methods.

## 3 Theoretical background

In this section, we describe the KL divergence (Section 3.2), the probability model of SGNS (Section 3.3), and the exponential family of distributions (Section 3.4) that are the background of our theoretical argument in the next section.

### 3.1 Preliminary

**Probability distributions.** We denote the probability of a word $w$ in the corpus as $p(w)$ and the unigram distribution of the corpus as $p(\cdot)$. Also, we denote the conditional probability of a word $w'$ co-occurring with $w$ within a fixed-width window as $p(w'|w)$, and the co-occurrence distribution as

$p(\cdot|w)$. Since these are probability distributions, $\sum_{w \in V} p(w) = \sum_{w' \in V} p(w'|w) = 1$, where $V$ is the vocabulary set of the corpus. The frequency-weighted average of $p(\cdot|w)$ is again the unigram distribution $p(\cdot)$, that is,

$$p(\cdot) = \sum_{w \in V} p(w)p(\cdot|w). \tag{1}$$

**Embeddings.** SGNS learns two different embeddings with dimensions $d$ for each word in $V$: word embedding $u_w \in \mathbb{R}^d$ for $w \in V$ and context embedding $v_{w'} \in \mathbb{R}^d$ for $w' \in V$. We denote the frequency-weighted averages of $u_w$ and $v_{w'}$ as

$$\bar{u} = \sum_{w \in V} p(w)u_w, \quad \bar{v} = \sum_{w' \in V} p(w')v_{w'}. \tag{2}$$

We also use the centered vectors

$$\hat{u}_w := u_w - \bar{u}, \quad \hat{v}_{w'} := v_{w'} - \bar{v}.$$

### 3.2 KL divergence measures information gain

The distributional semantics (Harris, 1954; Firth, 1957) suggests that "similar words will appear in similar contexts" (Brunila and LaViolette, 2022). This implies that the conditional probability distribution $p(\cdot|w)$ represents the meaning of a word $w$. The difference between $p(\cdot|w)$ and the marginal distribution $p(\cdot)$ can therefore capture the additional information obtained by observing $w$ in a corpus.

A metric for such discrepancies of information is the KL divergence of $p(\cdot|w)$ to $p(\cdot)$, defined as

$$\text{KL}(p(\cdot|w) \| p(\cdot)) = \sum_{w' \in V} p(w'|w) \log \frac{p(w'|w)}{p(w')}.$$

In this paper, we denote it by $\text{KL}(w)$ and call it the KL divergence of word $w$. Since $p(\cdot)$ is the prior distribution and $p(\cdot|w)$ is the posterior distribution given the word $w$, $\text{KL}(w)$ can be interpreted as the information gain of word $w$ (Oladyshkin and Nowak, 2019). Since the joint distribution of $w'$ and $w$ is $p(w', w) = p(w'|w)p(w)$, the expected value of $\text{KL}(w)$ is expressed as

$$\sum_{w \in V} p(w)\text{KL}(w)$$
$$= \sum_{w \in V} \sum_{w' \in V} p(w', w) \log \frac{p(w', w)}{p(w')p(w)}.$$

This is the mutual information $I(W', W)$ of the two random variables $W'$ and $W$ that correspond to $w'$ and $w$, respectively[2]. $I(W', W)$ is often called information gain in the literature.

---

[2]In the following, $w'$ and $w$ represent $W'$ and $W$ by abuse of notation.

### 3.3 The probability model of SGNS

The SGNS training utilizes the Noise Contrastive Estimation (NCE) (Gutmann and Hyvärinen, 2012) to distinguish between $p(\cdot|w)$ and the negative sampling distribution $q(\cdot) \propto p(\cdot)^{3/4}$. For each co-occurring word pair $(w, w')$ in the corpus, $\nu$ negative samples $\{w_i''\}_{i=1}^{\nu}$ are generated, and we aim to classify the $\nu+1$ samples $\{w', w_1'', \dots, w_\nu''\}$ as either a positive sample generated from $w' \sim p(w'|w)$ or a negative sample generated from $w'' \sim q(w'')$. The objective of SGNS (Mikolov et al., 2013) involves computing the probability of $w'$ being a positive sample using a kind of logistic regression model, which is expressed as follows (Gutmann and Hyvärinen, 2012):

$$\frac{p(w'|w)}{p(w'|w) + \nu q(w')} = \frac{1}{1 + e^{-\langle u_w, v_{w'} \rangle}}. \quad (3)$$

To gain a better understanding of this formula, we can cross-multiply both sides of (3) by the denominators:

$$p(w'|w)(1 + e^{-\langle u_w, v_{w'} \rangle}) = p(w'|w) + \nu q(w'),$$

and rearrange it to obtain:

$$p(w'|w) = \nu q(w') e^{\langle u_w, v_{w'} \rangle}. \quad (4)$$

We assume that the co-occurrence distribution satisfies the probability model (4). This is achieved when the word embeddings $\{u_w\}$ and $\{v_{w'}\}$ perfectly optimize the SGNS's objective, whereas it holds only approximately in reality.

### 3.4 Exponential family of distributions

We can generalize (4) by considering an instance of the exponential family of distributions (Lehmann and Casella, 1998; Barndorff-Nielsen, 2014; Efron, 2022), given by

$$p(w'|u) := q(w') \exp(\langle u, v_{w'} \rangle - \psi(u)), \quad (5)$$

where $u \in \mathbb{R}^d$ is referred to as the natural parameter vector, $v_{w'} \in \mathbb{R}^d$ represents the sufficient statistics (treated as constant vectors here, while tunable parameters in SGNS model), and the normalizing function is defined as

$$\psi(u) := \log \sum_{w' \in V} q(w') \exp(\langle u, v_{w'} \rangle),$$

ensuring that $\sum_{w' \in V} p(w'|u) = 1$ for any $u \in \mathbb{R}^d$. The SGNS model (4) is interpreted as a special case of the exponential family

$$p(w'|w) = p(w'|u_w)$$

for $u = u_w$ with constraints $\psi(u_w) = -\log \nu$ for $w \in V$; the model (5) is a curved exponential family when the parameter value $u$ is constrained as $\psi(u) = -\log \nu$, but we do not assume it in the following argument.

This section outlines some well-known basic properties of the exponential family of distributions, which have been established in the literature (Barndorff-Nielsen, 2014; Efron, 1978, 2022; Amari, 1982). For ease of reference, we provide the derivations of these basic properties in Appendix J.

The expectation and the covariance matrix of $v_{w'}$ with respect to $w' \sim p(w'|u)$ are calculated as the first and second derivatives of $\psi(u)$, respectively. Specifically, we have

$$\eta(u) := \frac{\partial \psi(u)}{\partial u} = \sum_{w' \in V} p(w'|u) v_{w'}, \quad (6)$$

$$G(u) := \frac{\partial^2 \psi(u)}{\partial u \partial u^\top} =$$
$$\sum_{w' \in V} p(w'|u)(v_{w'} - \eta(u))(v_{w'} - \eta(u))^\top. \quad (7)$$

The KL divergence of $p(\cdot|u_1)$ to $p(\cdot|u_2)$ for two parameter values $u_1, u_2 \in \mathbb{R}^d$ is expressed as

$$\mathrm{KL}(p(\cdot|u_1) \| p(\cdot|u_2)) =$$
$$\langle u_1 - u_2, \eta(u_1) \rangle - \psi(u_1) + \psi(u_2). \quad (8)$$

The KL divergence is interpreted as the squared distance between two parameter values when they are not very far from each other. In fact, the KL divergence (8) is expressed approximately as

$$2\mathrm{KL}(p(\cdot|u_1) \| p(\cdot|u_2))$$
$$\simeq (u_1 - u_2)^\top G(u_i)(u_1 - u_2) \quad (9)$$

for $i = 1, 2$. Here, the equation holds approximately by ignoring higher order terms of $O(\|u_1 - u_2\|^3)$. For more details, refer to Amari (1982, p. 369), Efron (2022, p. 35). More generally, $G(u)$ is the Fisher information metric, and (9) holds for a wide class of probability models (Amari, 1998).

## 4 Squared norm of word embedding approximates KL divergence

In this section, we theoretically explain the linear relationship between $\mathrm{KL}(w)$ and $\|u_w\|^2$ observed in Fig. 1 by elaborating on additional details of the exponential family of distributions (Section 4.1) and experimentally confirm our theoretical results (Section 4.2).

## 4.1 Derivation of theoretical results

We assume that the unigram distribution is represented by a parameter vector $u_0 \in \mathbb{R}^d$ and

$$p(w') = p(w'|u_0). \tag{10}$$

By substituting $u_1$ and $u_2$ with $u_w$ and $u_0$ respectively in (9), we obtain

$$2\mathrm{KL}(w) \simeq (u_w - u_0)^\top G (u_w - u_0). \tag{11}$$

Here $G := G(u_0)$ is the covariance matrix of $v_{w'}$ with respect to $w' \sim p(w')$, and we can easily compute it from (7) as

$$G = \sum_{w' \in V} p(w')(v_{w'} - \bar{v})(v_{w'} - \bar{v})^\top,$$

because $\eta(u_0) = \bar{v}$ from (2) and (6). However, it is important to note that the value of $u_0$ is not trained in practice, and thus we need an estimate of $u_0$ to compute $u_w - u_0$ on the right-hand side of (11).

We argue that $u_w - u_0$ in (11) can be replaced by $u_w - \bar{u} = \hat{u}_w$ so that

$$2\mathrm{KL}(w) \simeq \hat{u}_w^\top G \, \hat{u}_w. \tag{12}$$

For a formal derivation of (12), see Appendix K. Intuitively speaking, $\bar{u}$ approximates $u_0$, because $\bar{u}$ corresponds to $p(\cdot)$ in the sense that $\bar{u}$ is the weighted average of $u_w$ as seen in (2), while $p(\cdot)$ is the weighted average of $p(\cdot|u_w)$ as seen in (1).

To approximate $u_0$, we could also use $u_w$ of some representative words instead of using $\bar{u}$. We expect $u_0$ to be very close to some $u_w$ of stop-words such as '*a*' and '*the*' since their $p(\cdot|u_w)$ are expected to be very close to $p(\cdot)$.

Let us define a linear transform of the centered embedding as

$$\tilde{u}_w := G^{\frac{1}{2}} \hat{u}_w, \tag{13}$$

i.e., the *whitening of $u_w$ with the context embedding*[3] , then (12) is now expressed[4] as

$$2\mathrm{KL}(w) \simeq \|\tilde{u}_w\|^2. \tag{14}$$

Therefore, the square of the norm of the word embedding with the whitening-like transformation in (13) approximates the KL divergence.

---

[3]Note that the usual whightening is $\mathrm{Cov}(u)^{-\frac{1}{2}}\hat{u}_w$, but we call (13) as "whitening" for convenience in this paper.

[4](12) and (14) are equivalent, because $\|\tilde{u}_w\|^2 = \tilde{u}_w^\top \tilde{u}_w = (G^{\frac{1}{2}}\hat{u}_w)^\top G^{\frac{1}{2}}\hat{u}_w = \hat{u}_w^\top G^{\frac{1}{2}\top} G^{\frac{1}{2}}\hat{u}_w = \hat{u}_w^\top G\hat{u}_w$.

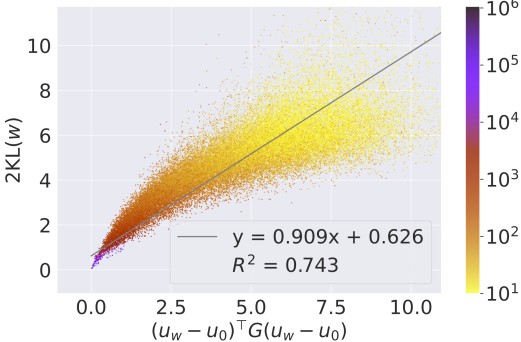

Figure 2: Confirmation of (11). The slope coefficient of 0.909, which is close to 1, indicates the validity of the theory.

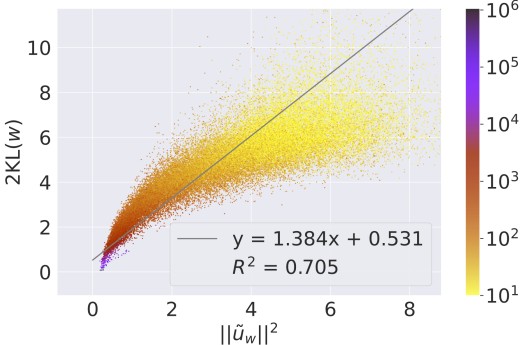

Figure 3: Confirmation of (12) and (14). The slope coefficient of 1.384, which is close to 1, suggests the validity of the theory.

## 4.2 Experimental confirmation of theory

The theory explained so far was confirmed by an experiment on real data.

**Settings.** We used the text8 corpus (Mahoney, 2011) with the size of $N = 17.0 \times 10^6$ tokens and $|V| = 254 \times 10^3$ vocabulary words. We trained 300-dimensional word embeddings $(u_w)_{w \in V}$ and $(v_{w'})_{w' \in V}$ by optimizing the objective of SGNS model (Mikolov et al., 2013). We also computed the KL divergence $(\mathrm{KL}(w))_{w \in V}$ from the co-occurrence matrix. These embeddings and KL divergence are used throughout the paper. See Appendix A for the details of the settings.

**Details of Fig. 1.** First, look at the plot of $\mathrm{KL}(w)$ and $\|u_w\|^2$ in Fig. 1 again. Although $u_w$ are raw word embeddings without the transformation (13), we confirm good linearity $\|u_w\|^2 \propto \mathrm{KL}(w)$. A regression line was fitted to words with $n_w > 10^3$, where low-frequency words were not very stable and ignored. The coefficient of determination $R^2 = 0.831$ indicates a very good fitting.

**Adequacy of theoretical assumptions.** In Fig. 1, the minimum value of $\mathrm{KL}(w)$ is observed to be very close to zero. This indicates that $p(\cdot|w)$ for the most frequent $w$ is very close to $p(\cdot)$ in the corpus, and that the assumption (10) in Section 4.1 is adequate.

**Confirmation of the theoretical results.** To confirm the theory stated in (11), we thus estimated $u_0$ as the frequency-weighted average of word vectors corresponding to the words {*the, of, and*}. These three words were selected as they are the top three words in the word frequency $n_w$. Then the correctness of (11) was verified in Fig. 2, where the slope coefficient is much closer to 1 than 0.048 of Fig. 1. Similarly, the fitting in Fig. 3 confirmed the theory stated in (12) and (14), where we replaced $u_0$ by $\bar{u}$.

**Experiments on other embeddings.** In Appendix G, the theory was verified by performing experiments using a larger corpus of Wikipedia dump (Wikimedia Foundation, 2021). In Appendix H, we also confirmed similar results using pre-trained fastText (Bojanowski et al., 2017) and SGNS (Li et al., 2017) embeddings.

# 5 Contextualized embeddings

The theory developed for static embeddings of the SGNS model is extended to contextualized embeddings in language models, or any neural networks with the softmax output layer.

## 5.1 Theory for language models

The final layer of language models with weights $v_{w'} \in \mathbb{R}^d$ and bias $b_{w'} \in \mathbb{R}$ is expressed for contextualized embedding $u \in \mathbb{R}^d$ as

$$y_{w'} = \langle u, v_{w'} \rangle + b_{w'},$$

and the probability of choosing the word $w' \in V$ is calculated by the softmax function

$$p_{\mathrm{softmax}}(w'|u) = \frac{e^{y_{w'}}}{\sum_{w \in V} e^{y_w}}. \qquad (15)$$

Comparing (15) with (5), the final layer is actually interpreted as the exponential family of distributions with $q(w') = e^{b_{w'}}/\sum_{w \in V} e^{b_w}$ so that $p_{\mathrm{softmax}}(w'|u) = p(w'|u)$. Thus, the theory for SGNS based on the exponential family of distributions should hold for language models.

However, we need the following modifications to interpret the theory. Rather than representing the co-occurrence distribution, $p(\cdot|u)$ now signifies the word distribution at a specific token position provided with the contextualized embedding $u$. Instead of the frequency-weighted average $\bar{u} = \sum_{w \in V} p(w)u_w$, we redefine $\bar{u} := \sum_{i=1}^{N} u_i/N$ as the average over the contextualized embeddings $\{u_i\}_{i=1}^{N}$ calculated from the training corpus of the language model. Here, $u_i$ denotes the contextualized embedding computed for the $i$-th token of the training set of size $N$. The information gain of contextualized embedding $u$ is

$$\mathrm{KL}(u) := \mathrm{KL}(p(\cdot|u) \parallel p(\cdot)).$$

With these modifications, all the arguments presented in Sections 3.4 and 4.1, along with their respective proofs, remain applicable in the same manner (Appendix L), and we have the main result (14) extended to contextualized embeddings as

$$2\mathrm{KL}(u) \simeq \|\tilde{u}\|^2, \qquad (16)$$

where the contextualized version of the centering and whitening are expressed as $\hat{u} := u - \bar{u}$ and $\tilde{u} := G^{\frac{1}{2}}\hat{u}$, respectively.

## 5.2 Experimental confirmation of theory

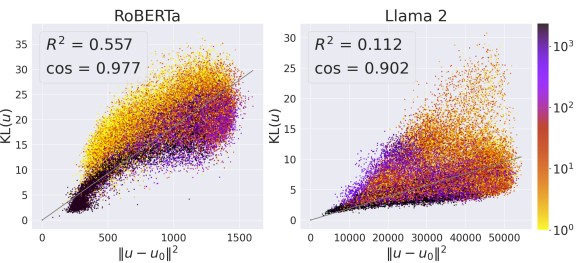

Figure 4: Linear relationship between the KL divergence and the squared norm of contextualized embedding for RoBERTa and Llama 2. The color represents token frequency.

We have tested four pre-trained language models: BERT (Devlin et al., 2019), RoBERTa (Liu et al., 2019), GPT-2 (Radford et al., 2019), and Llama 2 (Touvron et al., 2023) from Hugging Face transformers library (Wolf et al., 2020). Since the assumption (10) may not be appropriate for these models, we first computed $u_0 = \mathrm{argmin}_{u \in \{u_1,...,u_N\}}\mathrm{KL}(u)$, and used $p(\cdot|u_0)$ as a substitute for $p(\cdot)$ when verifying the linear relationship between $\mathrm{KL}(u)$ and $\|u - u_0\|^2$. Fig. 4 demonstrates that the linear relationship holds approximately for RoBERTa and Llama 2. All results,

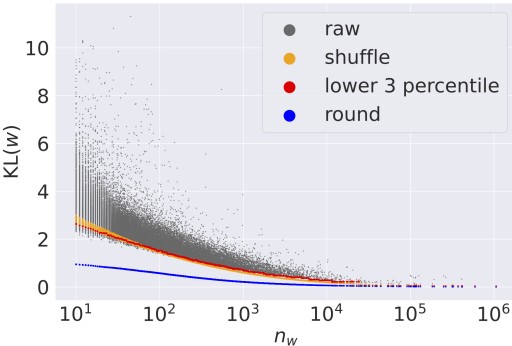

Figure 5: KL divergence computed with four different procedures plotted against word frequency $n_w$ for the same words in Fig. 1. 'raw', 'shuffle', and 'round' are $\mathrm{KL}(w)$, $\overline{\mathrm{KL}}(w)$, and $\mathrm{KL}_0(w)$, respectively. 'lower 3 percentile' is the lower 3-percentile point of $\mathrm{KL}(w)$ at each word frequency bin.

including those for BERT and GPT-2, as well as additional details, are described in Appendix I. While not as distinct as the result from SGNS in Fig. 1, it was observed that the theory suggested by (16) approximately holds true in the case of contextualized embeddings from language models.

## 6 Word frequency bias in KL divergence

The KL divergence is highly correlated with word frequency. In Fig. 5, 'raw' shows the plot of $\mathrm{KL}(w)$ against $n_w$. The KL divergence tends to be larger for less frequent words. A part of this tendency represents the true relationship that rarer words are more informative and thus tend to shift the co-occurrence distribution from the corpus distribution. However, a large part of the tendency, particularly for low-frequency words, comes from the error caused by the finite size $N$ of the corpus. This introduces a spurious relationship between $\mathrm{KL}(w)$ and $n_w$, causing a direct influence of word frequency. The word informativeness can be better measured by using the KL divergence when this error is adequately corrected.

### 6.1 Estimation of word frequency bias

**Preliminary.** The word distributions $p(\cdot)$ and $p(\cdot|w)$ are calculated from a finite-length corpus. The observed probability of a word $w$ is $p(w) = n_w/N$, where $N = \sum_{w \in V} n_w$. The observed probability of a context word $w'$ co-occurring with $w$ is $p(w'|w) = n_{w,w'}/\sum_{w'' \in V} n_{w,w''}$, where $(n_{w,w'})_{w,w' \in V}$ is the co-occurrence matrix. We computed $n_{w,w'}$ as the number of times that $w'$ appears within a window of $\pm h$ around $w$ in the corpus. Note that the denominator of $p(w'|w)$ is $\sum_{w'' \in V} n_{w,w''} = 2hn_w$ if the endpoints of the corpus are ignored.

**Sampling error ('shuffle').** Now we explain how word frequency directly influences the KL divergence. Consider a randomly shuffled corpus, i.e., words are randomly reordered from the original corpus (Montemurro and Zanette, 2010; Tanaka-Ishii, 2021). The unigram information, i.e., $n_w$ and $p(\cdot)$, remains unchanged after shuffling the corpus. On the other hand, the bigram information, i.e., $n_{w,w'}$ and $p(\cdot|w)$, computed for the shuffled corpus is independent of the co-occurrence of words in the original corpus. In the limit of $N \to \infty$, $p(\cdot|w) = p(\cdot)$ holds and $\mathrm{KL}(w) = 0$ for all $w \in V$ in the shuffled corpus. For finite corpus size $N$, however, $p(\cdot|w)$ deviates from $p(\cdot)$ because $(n_{w,w'})_{w' \in V}$ is approximately interpreted as a sample from the multinomial distribution with parameter $p(\cdot)$ and $2hn_w$.

In order to estimate the error caused by the direct influence of word frequency, we generated 10 sets of randomly shuffled corpus and computed the average of $\mathrm{KL}(w)$, denoted as $\overline{\mathrm{KL}}(w)$, which is shown as 'shuffle' in Fig. 5. $\overline{\mathrm{KL}}(w)$ does not convey the bigram information of the original corpus but does represent the sampling error of the multinomial distribution. For sufficiently large $N$, we expect $\overline{\mathrm{KL}}(w) \approx 0$ for all $w \in V$. However, $\overline{\mathrm{KL}}(w)$ is very large for small $n_w$ in Fig. 5.

**Sampling error ('lower 3 percentile').** Another computation of $\overline{\mathrm{KL}}(w)$ faster than 'shuffle' was also attempted as indicated as 'lower 3 percentile' in Fig. 5. This represents the lower 3-percentile point of $\mathrm{KL}(w)$ in a narrow bin of word frequency $n_w$. First, 200 bins were equally spaced on a logarithmic scale in the interval from 1 to $\max(n_w)$. Next, each bin was checked in order of decreasing $n_w$ and merged so that each bin had at least 50 data points. This method allows for faster and more robust computation of $\overline{\mathrm{KL}}(w)$ directly from $\mathrm{KL}(w)$ of the original corpus without the need for shuffling.

**Quantization error ('round').** There is another word frequency bias due to the fact that the co-occurrence matrix only takes integer values; it is indicated as 'round' in Fig. 5. This quantization error is included in the sampling error estimated by $\overline{\mathrm{KL}}(w)$, so there is no need for further correction. See Appendix C for details.

## 6.2 Correcting word frequency bias

We simply subtracted $\overline{\text{KL}}(w)$ from $\text{KL}(w)$. The sampling error $\overline{\text{KL}}(w)$ was estimated by either 'shuffle' or 'lower 3 percentile'. We call

$$\Delta\text{KL}(w) := \text{KL}(w) - \overline{\text{KL}}(w) \qquad (17)$$

as the bias-corrected KL divergence. The same idea using the random word shuffling has been applied to an entropy-like word statistic in an existing study (Montemurro and Zanette, 2010).

## 7 Experiments

In the experiments, we first confirmed that the KL divergence is indeed a good metric of the word informativeness (Section 7.1). Then we confirmed that the norm of word embedding encodes the word informativeness as well as the KL divergence (Section 7.2). Details of the experiments are given in Appenices D, E, and F.

As one of the baseline methods, we used the Shannon entropy of $p(\cdot|w)$, defined as

$$H(w) = - \sum_{w' \in V} p(w'|w) \log p(w'|w).$$

It also represents the information conveyed by $w$ as explained in Appendix B.

| Dataset | random | $n_w$ | $n_wH(w)$ | $n_w\text{KL}(w)$ |
|---|---|---|---|---|
| Krapivin2009 | 0.86 | 6.17 | 6.13 | **9.59** |
| theses100 | 0.97 | 9.69 | 9.79 | **12.31** |
| fao780 | 1.61 | 11.77 | 11.84 | **15.39** |
| SemEval2010 | 1.67 | 9.52 | 9.50 | **11.10** |
| Nguyen2007 | 1.90 | 10.56 | 10.57 | **12.84** |
| PubMed | 2.89 | 8.28 | 8.25 | **11.93** |
| citeulike180 | 4.01 | **18.20** | 18.18 | 17.98 |
| wiki20 | 4.15 | 9.32 | 9.23 | **19.90** |
| fao30 | 4.92 | 15.92 | 17.05 | **36.88** |
| Schutz2008 | 8.36 | 22.32 | **22.83** | 20.93 |
| kdd | 10.14 | **18.27** | 18.24 | 10.08 |
| Inspec | 10.54 | **16.31** | 16.22 | 14.61 |
| www | 12.08 | **21.20** | 21.11 | 12.76 |
| SemEval2017 | 14.16 | 19.86 | 19.62 | **20.85** |
| KPCrowd | 39.64 | 25.73 | 25.82 | **40.47** |

Table 2: MRR of keyword extraction experiment. For complete results on MRR and P@5, see Tables 6 and 7, respectively, in Appendix D.

## 7.1 KL divergence represents the word informativeness

Through keyword extraction tasks, we confirmed that the KL divergence is indeed a good metric of the word informativeness.

**Settings.** We used 15 public datasets for keyword extraction for English documents. Treating each document as a "corpus", vocabulary words were ordered by a measure of informativeness, and Mean Reciprocal Rank (MRR) was computed as an evaluation metric. When a keyword consists of two or more words, the worst value of rank was used. We used specific metrics, namely 'random', $n_w$, $n_wH(w)$ and $n_w\text{KL}(w)$, as our baselines. These metrics are computed only from each document without relying on external knowledge, such as a dictionary of stopwords or a set of other documents. For this reason, we did not use other metrics, such as TF-IDF, as our baselines. Note that $\|u_w\|^2$ was not included in this experiment because embeddings cannot be trained from a very short "corpus".

**Results and discussions.** Table 2 shows that $n_w\text{KL}(w)$ performed best in many datasets. Therefore, keywords tend to have a large value of $n_w\text{KL}(w)$, and thus $p(\cdot|w)$ is significantly different from $p(\cdot)$. This result verifies the idea that keywords have significantly large information gain.

## 7.2 Norm of word embedding encodes the word informativeness

We confirmed through proper-noun discrimination tasks (Section 7.2.1) and hypernym discrimination tasks (Section 7.2.2) that the norm of word embedding, as well as the KL divergence, encodes the word informativeness, and also confirmed that correcting the word frequency bias improves it.

In these experiments, we examined the properties of the raw word embedding $u_w$ instead of the whitening-like transformed word embedding $\tilde{u}_w$. From a practical standpoint, we used $u_w$, but experiments using $\tilde{u}_w$ exhibited a similar trend.

**Correcting word frequency bias.** In the same way as (17), we correct the bias of embedding norm and denote the bias-corrected squared norm as $\Delta\|u_w\|^2 := \|u_w\|^2 - \overline{\|u_w\|^2}$. We used the 'lower 3 percentile' method of Section 6.1 for $\Delta\|u_w\|^2$, because the recomputation of embeddings for the shuffled corpus is prohibitive. Other bias-corrected quantities, such as $\Delta\text{KL}(w)$ and $\Delta H(w)$, were computed from 10 sets of randomly shuffled corpus.

### 7.2.1 Proper-noun discrimination

**Settings.** We used 10561 proper nouns, 123 function words, 4771 verbs, and 2695 adjectives that appeared in the text8 corpus not less than 10 times.

|  | $n_w$ | $H(w)$ | $\mathrm{KL}(w)$ | $\|u_w\|^2$ | $\Delta H(w)$ | $\Delta\mathrm{KL}(w)$ | $\Delta\|u_w\|^2$ |
|---|---|---|---|---|---|---|---|
| proper nouns vs. verbs | 0.519 | 0.582 | 0.651 | 0.656 | 0.715 | 0.826 | **0.842** |
| proper nouns vs. adjectives | 0.543 | 0.581 | 0.613 | 0.626 | 0.645 | 0.699 | **0.728** |

Table 3: Binary classification of part-of-speech. Values are the ROC-AUC (higher is better). See Fig. 9 in Appendix E for histograms of measures.

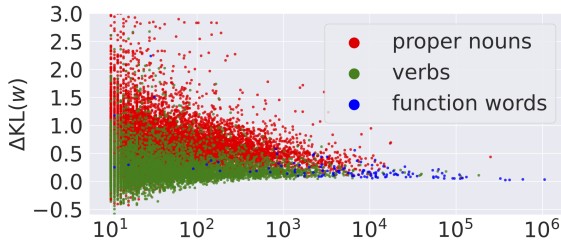 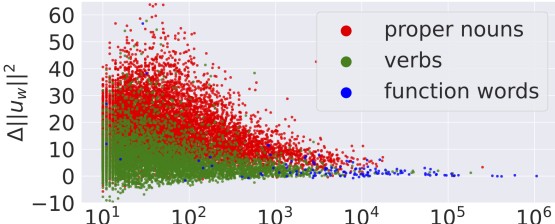

Figure 6: The bias-corrected KL divergence $\Delta\mathrm{KL}(w)$ and the bias-corrected squared norm of word embedding $\Delta\|u_w\|^2$ are plotted against word frequency $n_w$. Each dot represents a word; 10561 proper nouns (red dots), 123 function words (blue dots), and 4771 verbs (green dots). The same plot for adjectives, which is omitted in the figure, produced a scatterplot that almost overlapped with the verbs.

|  | $n_{\mathrm{hyper}}/n_{\mathrm{hypo}}$ | | |
|---|---|---|---|
|  | $> 1$ | $< 1$ | ave. |
| random | 50.00 | 50.00 | 50.00 |
| $n_w$ | 100.00 | 0.00 | 50.00 |
| WeedsPrec | 95.05 | 7.61 | 51.33 |
| SLQS Row | 95.20 | 13.70 | 54.45 |
| SLQS | 82.69 | 42.82 | 62.76 |
| $\mathrm{KL}(w)$ | 96.46 | 17.84 | 57.15 |
| $\|u_w\|^2$ | 94.07 | 24.89 | 59.48 |
| $\Delta$WeedsPrec | 46.53 | 51.88 | 49.20 |
| $\Delta$SLQS Row | 59.75 | 43.14 | 51.44 |
| $\Delta$SLQS | 50.41 | 69.06 | 59.74 |
| $\Delta\mathrm{KL}(w)$ | 65.90 | 62.94 | 64.42 |
| $\Delta\|u_w\|^2$ | 75.86 | 61.81 | **68.84** |

Table 4: Accuracy of hypernym-hyponym classification; the unweighted average over the four datasets. See Table 9 in Appendix F for the complete result.

We used $n_w$, $H(w)$, $\mathrm{KL}(w)$, and $\|u_w\|^2$ as a measure for discrimination. The performance of binary classification was evaluated by ROC-AUC.

**Results and discussions.** Table 3 shows that $\Delta\mathrm{KL}(w)$ and $\Delta\|u_w\|^2$ can discriminate proper nouns from other parts of speech more effectively than alternative measures. A larger value of $\Delta\mathrm{KL}(w)$ and $\Delta\|u_w\|^2$ indicates that words appear in a more limited context. Fig. 6 illustrates that proper nouns tend to have larger $\Delta\mathrm{KL}(w)$ and $\Delta\|u_w\|^2$ values when compared to verbs and function words.

### 7.2.2 Hypernym discrimination

**Settings.** We used English hypernym-hyponym pairs extracted from four benchmark datasets for hypernym discrimination: BLESS (Baroni and Lenci, 2011), EVALution (Santus et al., 2015), Lenci/Benotto (Lenci and Benotto, 2012), and Weeds (Weeds et al., 2014). Each dataset was divided into two parts by comparing $n_w$ of hypernym and hyponym to remove the effect of word frequency. In addition to 'random' and $n_w$, we used WeedsPrec (Weeds and Weir, 2003; Weeds et al., 2004), SLQS Row (Shwartz et al., 2017) and SLQS (Santus et al., 2014) as baselines.

**Results and discussions.** Table 4 shows that $\Delta\|u_w\|^2$ and $\Delta\mathrm{KL}(w)$ were the best and the second best, respectively, for predicting hypernym in hypernym-hyponym pairs. Correcting frequency bias remedies the difficulty of discrimination for the $n_{\mathrm{hyper}} < n_{\mathrm{hypo}}$ part, resulting in an improvement in the average accuracy.

## 8 Conclusion

We showed theoretically and empirically that the KL divergence, i.e., the information gain of the word, is encoded in the norm of word embedding. The KL divergence and, thus, the norm of word embedding has the word frequency bias, which was corrected in the experiments. We then confirmed that the KL divergence and the norm of word embedding work as a metric of informativeness in NLP tasks.

## Limitations

- The important limitation of the paper is that the theory assumes the skip-gram with negative sampling (SGNS) model for static word embeddings or the softmax function in the final layer of language models for contextualized word embeddings.

- The theory also assumes that the model is trained perfectly, as mentioned in Section 3.3. When the assumption is violated, the theory may not hold. For example, the training is not perfect when the number of epochs is insufficient, as illustrated in Appendix G.

## Ethics Statement

This study complies with the ACL Ethics Policy[5].

## Acknowledgements

We would like to thank Junya Honda and Yoichi Ishibashi for the discussion and the anonymous reviewers for their helpful advice. This study was partially supported by JSPS KAKENHI 22H05106, 23H03355, JST ACT-X JPMJAX200S, and JST CREST JPMJCR21N3.

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

| Dimensionality | 300 |
|---|---|
| Epochs | 100 |
| Window size $h$ | 10 |
| Negative samples $\nu$ | 5 |
| Learning rate | 0.025 |
| Min count | 1 |

Table 5: SGNS parameters.

Clara Ma, Yacine Jernite, Julien Plu, Canwen Xu, Teven Le Scao, Sylvain Gugger, Mariama Drame, Quentin Lhoest, and Alexander Rush. 2020. Transformers: State-of-the-art natural language processing. In *Proceedings of the 2020 Conference on Empirical Methods in Natural Language Processing: System Demonstrations*, pages 38–45, Online. Association for Computational Linguistics.

Sho Yokoi, Ryo Takahashi, Reina Akama, Jun Suzuki, and Kentaro Inui. 2020. Word rotator's distance. In *Proceedings of the 2020 Conference on Empirical Methods in Natural Language Processing (EMNLP)*, pages 2944–2960, Online. Association for Computational Linguistics.

## A  Settings for computation of word embeddings and KL divergence

**Corpus.** We used the text8 (Mahoney, 2011), which is an English corpus data with the size of $N = 17.0 \times 10^6$ tokens and $|V| = 254 \times 10^3$ vocabulary words. We used all the tokens[6] separated by spaces for word embeddings and KL divergence.

**Training of the SGNS model.** Word embeddings were trained[7] by optimizing the same objective function used in Mikolov et al. (2013). Parameters used to train SGNS are summarized in Table 5. The learning rate shown is the initial value, which we decreased linearly to the minimum value of $1.0 \times 10^{-4}$ during the learning process. The negative sampling distribution was specified as

$$q(w) \propto (n_w)^{\frac{3}{4}}.$$

The elements of $u_w$ were initialized by the uniform distribution over $[-0.5, 0.5]$ divided by the dimensionality of the embedding, and the elements of $v_w$ were initialized by zero.

---

[6]We manually checked that the words used in Table 1 and Table 8 were not personally identifiable or offensive.

[7]We used AMD EPYC 7702 64-Core Processor (64 cores × 2). In this setting, the CPU time is estimated at about 12 hours.

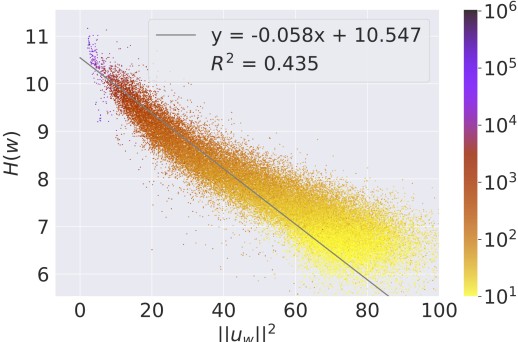

Figure 7: The Shannon entropy and the squared norm of word embedding. Settings are the same as in Fig. 1.

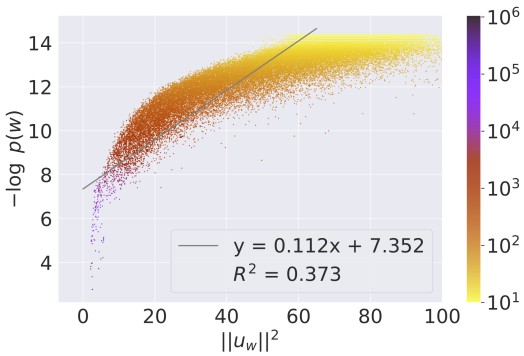

Figure 8: The self-information and the squared norm of word embedding. Settings are the same as in Fig. 1.

**Computation of KL divergence.** The value of $\mathrm{KL}(w)$ was computed from $p(\cdot|w)$ and $p(\cdot)$ using the definition in Section 3.2 with the convention that $0 \log 0 = 0$. The word probability $p(w')$ and the co-occurrence probability $p(w'|w)$ were computed from the word frequency $n_w$ and the co-occurrence matrix $(n_{w,w'})_{w,w' \in V}$, respectively, as described in Section 6. The co-occurrence matrix was computed with the window size $h = 10$.

**Word set for visualization.** We have used $47 \times 10^3$ words with $n_w \geq 10^1$ for the plots of Figs. 1 to 5. Except for Fig. 5, extreme points, up to 0.5% for each axis, were truncated to set the plot range. Word embeddings and KL divergence are not very stable for low-frequency words. For this reason, we used 1820 words with $n_w > 10^3$ to fit the simple linear regression model using the least squares method.

## B  Other quantities of information theory

In addition to KL divergence, two other information theoretic quantities are discussed here.

## B.1 Shannon entropy

The Shannon entropy of $p(\cdot|w)$, defined as

$$H(w) = -\sum_{w' \in V} p(w'|w) \log p(w'|w),$$

also represents information conveyed by $w$. In this paper, we call it the Shannon entropy of word $w$. $H(w)$ is closely related to $\mathrm{KL}(w)$. The Shannon entropy of $p(\cdot|w)$ can be written as

$$H(w) = \log |V| - \mathrm{KL}(p(\cdot|w) \parallel \mathrm{unif}(\cdot)),$$

meaning that $-H(w)$ measures how much the co-occurrence distribution shifts from the uniform distribution (i.e., $\mathrm{unif}(w') = 1/|V|$). Thus, $H(w)$ and $\mathrm{KL}(w)$ have different reference distributions.

## B.2 Self-information

A much naive way of measuring the information of a word is the self-information of the event that the word $w$ is sampled from $p(\cdot)$, defined as

$$I(w) = -\log p(w).$$

The expected value $\sum_{w \in V} p(w)I(w)$ is the Shannon entropy of $p(\cdot)$. Since $p(w)$ was computed as $p(w) = n_w/N$,

$$I(w) = \log N - \log n_w$$

actually looks at the word frequency $n_w$ in the log scale.

## B.3 Relation to word embedding

$H(w)$ and $I(w)$ were computed with the same settings as in Section 4.2 and Appendix A. They were plotted along with $\|u_w\|^2$ as shown in Fig. 7 and Fig. 8, respectively. Compared with $\mathrm{KL}(w)$, the relationships are less clear with $R^2 \approx 0.4$. From this experiment, we see that $\mathrm{KL}(w)$ better represents $\|u_w\|^2$ than $H(w)$ and $I(w)$.

## C Quantization error

The co-occurrence matrix $(n_{w,w'})_{w,w' \in V}$ is sparse with many zero values at rows of $w$ with small $n_w$. The effect of quantization error caused by $n_{w,w'}$ taking only integer values cannot be ignored for low-frequency words. This effect is part of the sampling error, but we try to isolate the quantization error here. Let us redefine $n_{w,w'} := \mathrm{round}(2hn_w p(w'))$ and compute the KL divergence, denoted as $\mathrm{KL}_0(w)$, which is shown

as 'round' in Fig. 5. If there is no rounding errors, $p(w'|w) = p(w')$ so that $\mathrm{KL}_0(w) = 0$. In reality, however, $\mathrm{KL}_0(w)$ is non-negligible for words with small $n_w$, and this effect can be corrected by $\mathrm{KL}(w) - \mathrm{KL}_0(w)$.

## D Details of experiment in Section 7.1

In this experiment, we confirmed that human-annotated keywords of documents were observed at the top of the ranking calculated by the discrepancy between $p(\cdot|w)$ and $p(\cdot)$.

**Datasets.** For the experiment of keyword extraction, we used 15 datasets in English[8]. Each entry consists of a pair of document and gold keywords. Table 6 includes information on the size (the number of documents) and the type of documents.

**Preparation.** Each document in the datasets was tokenized by NLTK's word_tokenize function. Then, each word was stemmed using NLTK's PorterStemmer, and all characters were converted to lowercase. The same preprocessing of stemming and lowercase was also applied to the gold keywords. However, we did *not* remove stopwords in preprocessing to see if the informativeness measures could remove unnecessary stopwords by themselves. The co-occurrence matrix for each document was computed with the window size $h = 10$. Note that only a subset $V' \subset V$ of the vocabulary set described below was used for stable computation of $p(w'|w)$, $w' \in V', w \in V$. For constructing $V'$, all the words $w \in V$ were sorted in decreasing order of $n_w$, and the cumulative frequency $c_i = \sum_{j=1}^{i} n_{w_j}$ up to the $i$-th frequent word were computed for $i = 1, 2, \ldots, |V|$. Then $V' = \{w_1, \ldots, w_i\}$ was defined with the smallest $i$ such that $c_i \geq N/3$.

**Methods.** In each document, word ranking lists were created by sorting its vocabulary words using the informativeness measures. For 'random', the

[8]Datasets for the keyword extraction experiment were obtained from a public repository https://github.com/LIAAD/KeywordExtractor-Datasets which includes Krapivin2009 (Krapivin et al., 2009), theses100 (Medelyan, 2015), fao780 and fao30 (Medelyan and Witten, 2008), SemEval2010 (Kim et al., 2010), Nguyen2007 (Nguyen and Kan, 2007), PubMed (Aronson et al., 2000), citeulike180 (Medelyan et al., 2009), wiki20 (Medelyan et al., 2008), Schutz2008 (Schutz, 2008), kdd (Gollapalli and Caragea, 2014), Inspec (Hulth, 2003), www (Gollapalli and Caragea, 2014), SemEval2017 (Augenstein et al., 2017), and KPCrowd (Marujo et al., 2011).

| Dataset | Size | Type | random | $n_w$ | $n_w H(w)$ | $\chi^2(w)$ | $n_w \text{KL}(w)$ |
|---|---|---|---|---|---|---|---|
| Krapivin2009 | 2304 | article | 0.86 | 6.17 | 6.13 | 8.00 | **9.59** |
| theses100 | 100 | article | 0.97 | 9.69 | 9.79 | 9.31 | **12.31** |
| fao780 | 779 | article | 1.61 | 11.77 | 11.84 | 11.04 | **15.39** |
| SemEval2010 | 243 | article | 1.67 | 9.52 | 9.50 | 8.40 | **11.10** |
| Nguyen2007 | 209 | article | 1.90 | 10.56 | 10.57 | 9.78 | **12.84** |
| PubMed | 500 | article | 2.89 | 8.28 | 8.25 | 9.91 | **11.93** |
| citeulike180 | 183 | article | 4.01 | **18.20** | 18.18 | 10.03 | 17.98 |
| wiki20 | 20 | report | 4.15 | 9.32 | 9.23 | 12.82 | **19.90** |
| fao30 | 30 | article | 4.92 | 15.92 | 17.05 | 29.47 | **36.88** |
| Schutz2008 | 1231 | article | 8.36 | 22.32 | **22.83** | 13.14 | 20.93 |
| kdd | 755 | abstract | 10.14 | **18.27** | 18.24 | 9.71 | 10.08 |
| Inspec | 2000 | abstract | 10.54 | **16.31** | 16.22 | 13.75 | 14.61 |
| www | 1330 | abstract | 12.08 | **21.20** | 21.11 | 11.67 | 12.76 |
| SemEval2017 | 493 | paragraph | 14.16 | 19.86 | 19.62 | 19.18 | **20.85** |
| KPCrowd | 500 | news | 39.64 | 25.73 | 25.82 | 39.02 | **40.47** |

Table 6: MRR of keyword extraction experiment.

ranking list is simply a random shuffle of the vocabulary words. For $n_w H(w)$, words were ranked in increasing order. For other measures, words were ranked in decreasing order. We multiply $n_w$ to $\text{KL}(w)$ because $G^2 = 2n_w \text{KL}(w)$ is appropriate for testing the null hypothesis that $p(\cdot|w) = p(\cdot)$. $n_w H(w)$ is also interpreted as a test statistic for testing the null hypothesis that $p(\cdot|w) = \text{unif}(\cdot)$. We also included the $\chi^2$ statistic (Matsuo and Ishizuka, 2004), which is related to $\text{KL}(w)$ as $\chi^2 \approx G^2$ for sufficiently large $n_w$.

**Evaluation metrics.** We used MRR and P@5 as evaluation metrics for the keyword prediction task.

**MRR** is the average of the reciprocals of gold keywords' ranks. The numbers in the tables were multiplied by 100. For each document, we used the best-ranked keyword, i.e., the minimum value of the ranks of correct answers. If a keyword is given as a phrase consisting of two or more words, the rank of the keyword is defined by the worst-ranked word. For example, the rank of "New York" is 10 if the ranks of "new" and "york" are 3 and 10, respectively.

**P@5** is the average percentage of correct answers that appear in the top five words of the ranked list. For each document, the number of gold keywords in the top five words was computed and divided by 5. For a keyword consisting of two or more words, it is regarded as a correct answer only when all the words are included in the top five words. Thus the percentage can be larger than 100

if several gold keywords share the same words.

**Results.** Table 6 shows MRR, and Table 7 shows P@5 of the experiment. Datasets were sorted in the increasing order of MRR of the random baseline in both tables. Table 2 in Section 7.1 is a summary of Table 6. Small values of MRR or P@5 of the random baseline indicate the extent of difficulty of the keyword extraction. Datasets with the article type are difficult, and the dataset with the news type is the easiest. In the difficult datasets, $n_w \text{KL}(w)$ performed best in almost all datasets.

# E    Details of experiment in Section 7.2.1

In this experiment, we confirmed that proper nouns tend to have larger values of $\Delta \text{KL}(w)$ and $\Delta \|u_w\|$ compared to other parts of speech.

**Datasets.** We used $10561$ proper nouns, $123$ function words, $4771$ verbs, and $2695$ adjectives that appeared in the text8 corpus not less than 10 times ($n_w \geq 10$). The parts of speech of these words were identified by NLTK's POS tagger. Proper nouns are tagged as {NN, NNS}, verbs are tagged as {VB, VBD, VBG, VBN, VBP, VBZ}, adjectives are tagged as {JJ, JJS, JJR}, and function words are tagged as {IN, PRP, PRP\$, WP, WP\$, DT, PDT, WDT, CC, MD, RP}. Proper nouns were restricted to those found in the $61711$ words of the English Proper nouns database[9].

[9] https://github.com/jxlwqq/english-proper-nouns/

| Dataset | Size | Type | random | $n_w$ | $n_w H(w)$ | $\chi^2(w)$ | $n_w \mathrm{KL}(w)$ |
|---|---|---|---|---|---|---|---|
| Krapivin2009 | 2304 | article | 0.11 | 0.80 | 0.83 | 2.37 | **3.12** |
| theses100 | 100 | article | 0.16 | 3.40 | 3.60 | 3.80 | **5.40** |
| fao780 | 779 | article | 0.28 | 3.70 | 3.72 | 3.75 | **5.52** |
| SemEval2010 | 243 | article | 0.23 | 1.89 | 1.81 | 2.63 | **4.28** |
| Nguyen2007 | 209 | article | 0.42 | 3.44 | 3.54 | 4.40 | **5.74** |
| PubMed | 500 | article | 0.54 | 2.08 | 2.00 | 2.96 | **3.76** |
| citeulike180 | 183 | article | 0.90 | **12.02** | 11.69 | 4.37 | 8.52 |
| wiki20 | 20 | report | 0.70 | 1.00 | 1.00 | 7.00 | **10.00** |
| fao30 | 30 | article | 1.53 | 9.33 | 8.67 | 14.67 | **18.00** |
| Schutz2008 | 1231 | article | 2.37 | 14.77 | **15.22** | 5.20 | 10.93 |
| kdd | 755 | abstract | 3.07 | 8.98 | **9.14** | 2.12 | 2.28 |
| Inspec | 2000 | abstract | 2.84 | **7.32** | 6.85 | 5.09 | 5.68 |
| www | 1330 | abstract | 3.78 | **10.98** | 10.89 | 2.33 | 3.07 |
| SemEval2017 | 493 | paragraph | 4.10 | **13.35** | 12.78 | 8.88 | 9.33 |
| KPCrowd | 500 | news | 21.75 | 18.37 | 18.33 | 21.25 | **24.33** |

Table 7: P@5 of keyword extraction experiment.

| $\Delta\mathrm{KL}(w)$ | Word Examples |
|---|---|
| Top (0% ∼ 10%) | HONDA, INTERPOL, Gabon, Yin, VAR, IMF, Benin, BO, Bene, GB |
| Middle (45% ∼ 55%) | Pete, Dee, Wine, Tony, Bogart, Alice, Cliff, Madonna, Dover, Leopold |
| Bottom (90% ∼ 100%) | storm, haven, sale, miracle, discover, Phillip, duty, prohibition, capitol, comfort |

Table 8: Randomly sampled proper nouns for each range of informativeness measured by the KL divergence.

**Preparation.** We computed $n_w$, $\mathrm{KL}(w)$ and $\|u_w\|^2$ from the text8 corpus as described in Appendix A. $H(w)$ was also computed in the same way as $\mathrm{KL}(w)$. For their bias-corrected versions, we used the 'shuffle' method in Section 6.1 for $\Delta\mathrm{KL}(w)$ and $\Delta H(w)$, and the 'lower 3 percentile' method for $\Delta\|u_w\|^2$. We used these measures for the binary classification of part-of-speech.

**Methods.** Proper nouns tend to have large values of $n_w$, $\mathrm{KL}(w)$ and $\|u_w\|^2$, or small values of $H(w)$ as seen in Fig. 9. Therefore, each word is classified as a proper noun if a measure is larger (or smaller) than a threshold value. We performed two sets of binary classification experiments: proper nouns vs. verbs, and proper nouns vs. adjectives.

**Evaluation metrics.** Since the classification depends on the threshold value, we used ROC-AUC to evaluate the classification performance. ROC-AUC was computed by Scipy's roc_curve function.

**Results.** Table 3 in Section 7.2.1 shows the ROC-AUC of the classification task, confirming the good performance of $\Delta\mathrm{KL}(w)$ and $\Delta\|u_w\|^2$.

Table 8 shows randomly sampled proper nouns with $10^1 \leq n_w \leq 10^3$ and specific ranges of $\Delta\mathrm{KL}(w)$; since our experiment is case-insensitive, some selected words were actually considered as common nouns, such as *storm* and *haven*. We observed that common nouns tend to have small KL values. On the other hand, words with large KL values include context-specific nouns, such as company names, suggesting that they are more informative.

## F   Details of experiment in Section 7.2.2

In this experiment, we confirmed that $\Delta\mathrm{KL}(w)$ and $\Delta\|u_w\|^2$ tend to have a smaller value for hypernym in hypernym-hyponym pairs.

**Datasets.** Among the hypernym-hyponym pairs in each dataset, we used those consisting of words that appear in the text8 corpus. Specifically, we used 1336 pairs from the 1337 pairs of the BLESS dataset (Baroni and Lenci, 2011), 3635 pairs from the 3637 pairs of the EVALution dataset (Santus et al., 2015), 1760 pairs from the 1933 pairs of the Lenci/Benotto dataset (Lenci and Benotto,

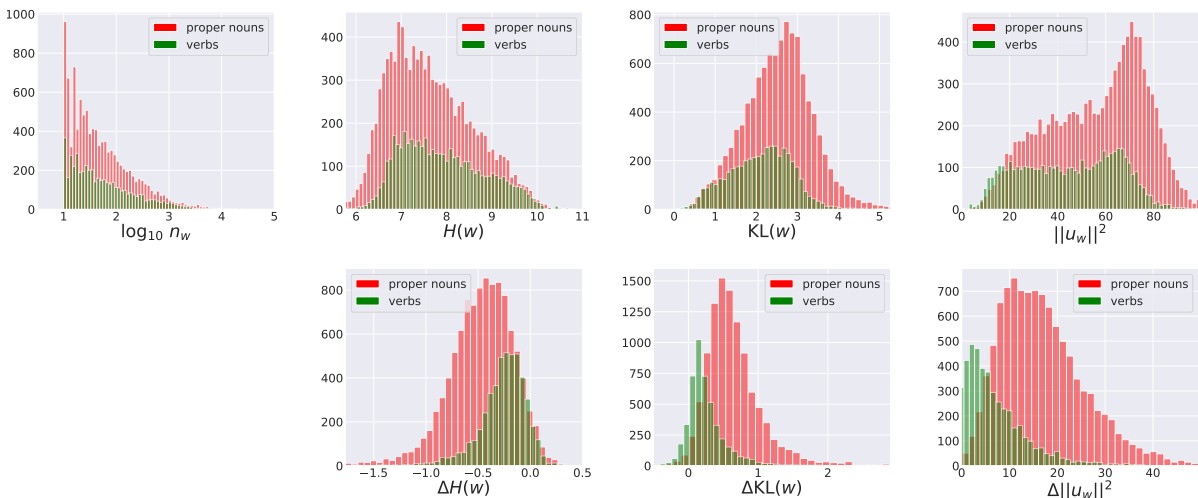

Figure 9: Histogram of each measure used for binary classification of part-of-speech. Plotted for 10561 proper nouns (red) and 4771 verbs (green) in the text8 corpus.

2012), 1427 pairs from the 1427 pairs of the Weeds dataset (Weeds et al., 2014). Each dataset was divided into two parts: the $n_{\text{hyper}} > n_{\text{hypo}}$ part and the $n_{\text{hyper}} < n_{\text{hypo}}$ part.

**Preparation.** We computed $n_w$, $n_{w,w'}$, $H(w)$, $KL(w)$, $\|u_w\|^2$, $\Delta H(w)$, $\Delta KL(w)$, and $\Delta\|u_w\|^2$ from the text8 corpus as described in Appendices A and E.

**Methods.** We considered the binary classification of hypernym given a hypernym-hyponym pair. Using $KL(w)$, $\|u_w\|^2$, $\Delta KL(w)$, or $\Delta\|u_w\|^2$ as a measure of informativeness, the word with a smaller value of the measure was predicted as hypernym.

Baseline methods to predict hypernym given a word pair $(w_1, w_2)$ are described below.

- **Random** is the random classification. The accuracy is 50%.

- **Word Frequency** chooses the word with larger $n_w$ as hypernym.

- **WeedsPrec** (Weeds and Weir, 2003; Weeds et al., 2004) is based on the distributional inclusion hypothesis that the context of hyponym is included in the context of its hypernym. The weighted inclusion of word $w_2$ in the context of word $w_1$ is formulated as

$$\text{WeedsPrec}(w_1, w_2) = \frac{\sum_{w' \in V_{w_1 \cap w_2}} n_{w_1, w'}}{\sum_{w' \in V} n_{w_1, w'}},$$

where $V_{w_1 \cap w_2} = \{w' \in V \mid n_{w_1, w'} > 0 \wedge n_{w_2, w'} > 0\}$. $w_1$ is predicted as hypernym if

$$\text{WeedsPrec}(w_1, w_2) < \text{WeedsPrec}(w_2, w_1).$$

- **SLQS Row** (Shwartz et al., 2017) compares the Shannon entropy. $w_1$ is predicted as hypernym if

$$SLQS_{Row}(w_1, w_2) := 1 - \frac{H(w_1)}{H(w_2)} < 0,$$

or equivalently $H(w_1) > H(w_2)$.

- **SLQS** (Santus et al., 2014) compares the median entropy of context words defined as

$$E(w) = \text{Median}_{c \in C_w} H(c).$$

$w_1$ is predicted as hypernym if

$$SLQS(w_1, w_2) := 1 - \frac{E(w_1)}{E(w_2)} < 0,$$

or equivalently $E(w_1) > E(w_2)$. Note that $C_w$ is the set of most strongly associated context words of $w$, as determined by positive local mutual information (Evert, 2005). We used $|C_w| = 50$.

- **ΔWeedsPrec** is the bias-corrected version of WeedsPrec computed by the method in Section 6.2. $\overline{\text{WeedsPrec}}(w_1, w_2)$ is the average of WeedsPrec$(w_1, w_2)$ for 10 randomly shuffled corpora, and $\Delta\text{WeedsPrec}(w_1, w_2) =$

| | $n_\text{hyper} > n_\text{hypo}$ | | | | $n_\text{hyper} < n_\text{hypo}$ | | | | |
|---|---|---|---|---|---|---|---|---|---|
| | BLESS | EVAL | LB | Weeds | BLESS | EVAL | LB | Weeds | average |
| size | 763 | 2394 | 1324 | 1022 | 573 | 1241 | 436 | 405 | |
| random | 50.00 | 50.00 | 50.00 | 50.00 | 50.00 | 50.00 | 50.00 | 50.00 | 50.00 |
| frequency | 100.00 | 100.00 | 100.00 | 100.00 | 0.00 | 0.00 | 0.00 | 0.00 | 50.00 |
| WeedsPrec | 93.97 | **94.78** | 96.45 | 95.01 | 4.54 | 8.30 | 8.49 | 9.14 | 51.33 |
| SLQS Row | 96.46 | 91.73 | 96.60 | **95.99** | 7.68 | 21.19 | 12.84 | 13.09 | 54.45 |
| SLQS | 87.94 | 84.04 | 83.16 | 75.64 | 52.53 | 46.25 | 40.14 | 32.35 | 62.76 |
| $KL(w)$ | **98.43** | 94.74 | **96.98** | 95.69 | 16.93 | 21.11 | 16.51 | 16.79 | 57.15 |
| $\|u_w\|^2$ | 98.17 | 93.69 | 94.49 | 89.92 | 28.27 | 27.56 | 22.25 | 21.48 | 59.48 |
| $\Delta$WeedsPrec | 35.78 | 46.07 | 50.83 | 53.42 | 57.77 | 49.88 | 50.00 | 49.88 | 49.20 |
| $\Delta$SLQS Row | 57.54 | 59.19 | 58.08 | 64.19 | 47.64 | 40.21 | 41.74 | 42.96 | 51.44 |
| $\Delta$SLQS | 55.83 | 55.93 | 50.45 | 39.43 | 73.30 | **66.00** | **72.25** | **64.69** | 59.74 |
| $\Delta KL(w)$ | 84.80 | 71.39 | 58.61 | 48.83 | 71.38 | 56.16 | 61.24 | 62.96 | 64.42 |
| $\Delta\|u_w\|^2$ | 91.87 | 75.23 | 72.73 | 63.60 | **74.69** | 58.26 | 55.05 | 59.26 | **68.84** |

Table 9: Accuracy of hypernym classification. For each method, $\Delta$Method is the bias-corrected version. We divided each dataset into two parts based on the word frequencies of hypernym ($n_\text{hyper}$) and hyponym ($n_\text{hypo}$). Dataset EVAL denotes EVALution.

$\text{WeedsPrec}(w_1, w_2) - \overline{\text{WeedsPrec}}(w_1, w_2)$. $w_1$ is predicted as hypernym if

$$\Delta\text{WeedsPrec}(w_1, w_2)$$
$$< \Delta\text{WeedsPrec}(w_2, w_1).$$

- **$\Delta$SLQS Row** is the bias-corrected version of SLQS Row. $w_1$ is predicted as hypernym if $\Delta H(w_1) > \Delta H(w_2)$.

- **$\Delta$SLQS** is the bias-corrected version of SLQS. $w_1$ is predicted as hypernym if $\Delta E(w_1) > \Delta E(w_2)$, where

$$\Delta E(w) = \text{Median}_{c \in C_w} \Delta H(c).$$

**Evaluation metrics.** The classification accuracy of each method was computed separately for the $n_\text{hyper} > n_\text{hypo}$ part and for the $n_\text{hyper} < n_\text{hypo}$ part of each dataset. Then, we calculated the unweighted average of the accuracy over the four datasets for each part and for both parts.

**Results.** Table 9 shows the classification accuracy. Table 4 in Section 7.2.2 is a summary of Table 9. Looking at the overall accuracy, $\Delta\|u_w\|^2$ and $\Delta KL(w)$ were the best and the second best, respectively, for predicting hypernym in hypernym-hyponym pairs.

# G   Results on Wikipedia dump

We used the Wikipedia dump (Wikimedia Foundation, 2021)[10] with the size of $N = 24.0 \times 10^8$ tokens and $|V| = 645 \times 10^4$ vocabulary words, which was preprocessed by Wikiextractor (Attardi, 2015). The training of the SGNS model and the computation of KL divergence were performed as in Appendix A using the same setting[11]. For plotting the results, we used 50,000 words randomly sampled from the 1,114,207 vocabulary words with $n_w \geq 10$. For fitting the regression line, we used 2,662 words with $n_w > 10^3$.

Fig. 10 shows the word embeddings of the Wikipedia dump computed with the same setting as that of the text8 corpus. The left panel of Fig. 10 is very similar to Fig. 1, confirming that the result for the text8 corpus is reproduced for the Wikipedia dump. The right panel of Fig. 10 corresponds to Fig. 8 with the axes exchanged and the $\log_{10} n_w$ axis rescaled. Again, the two plots are very similar.

However, the result changes when the epoch of training is reduced, thus the optimization is insufficient. Fig. 11 shows the word embeddings of the Wikipedia dump, but the epoch was reduced to 10.

---

[10]Wikipedia dump dataset is licensed under the GFDL and the CC BY-SA 3.0.

[11]We used AMD EPYC 7763 (64 cores). For 10 epochs of training, the CPU time is estimated at about 20 hours, and for 100 epochs of training, the CPU time is estimated at about 8 days.

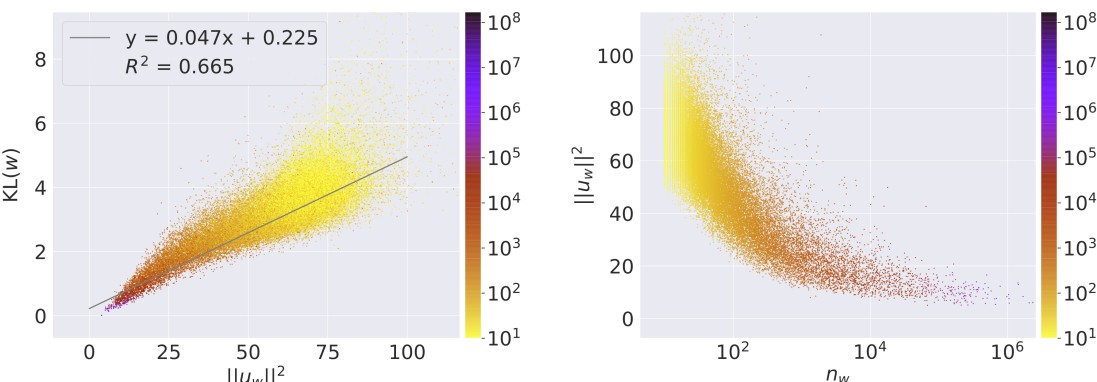

Figure 10: Word embeddings of Wikipedia dump computed with 100 epochs.

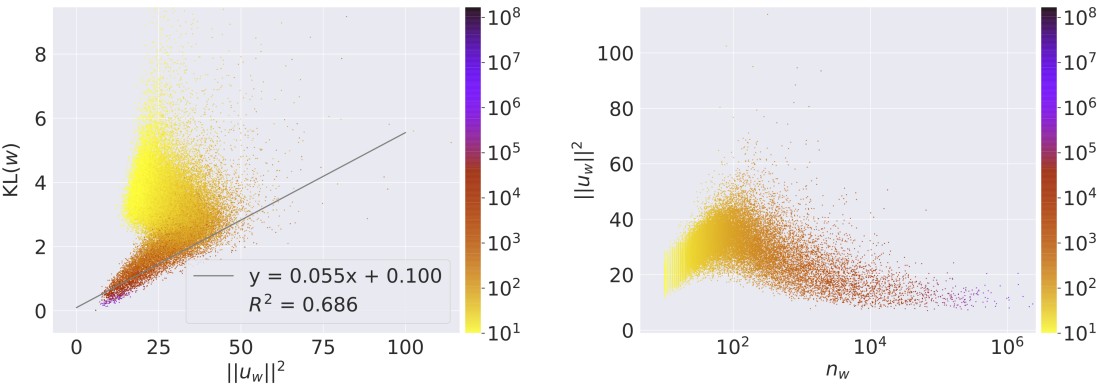

Figure 11: Word embeddings of Wikipedia dump computed with 10 epochs.

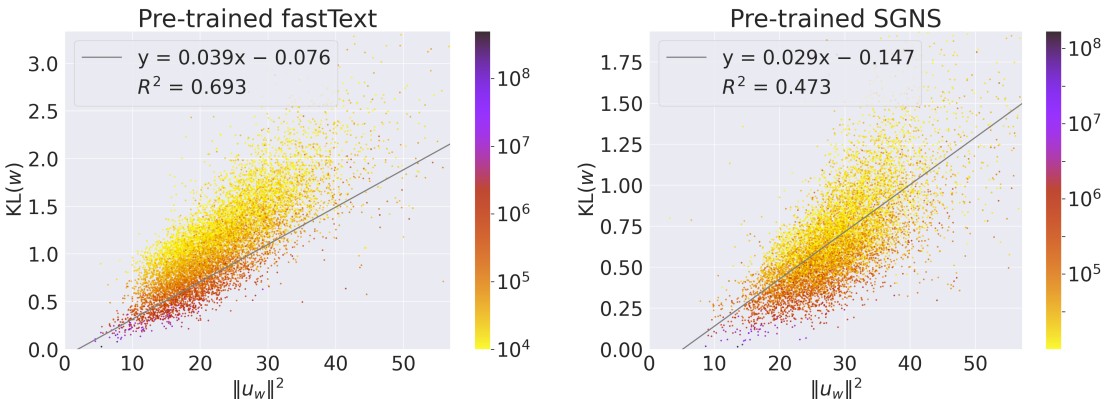

Figure 12: Two pre-trained word embeddings. Each regression line was fitted to all the points in the scatterplot.

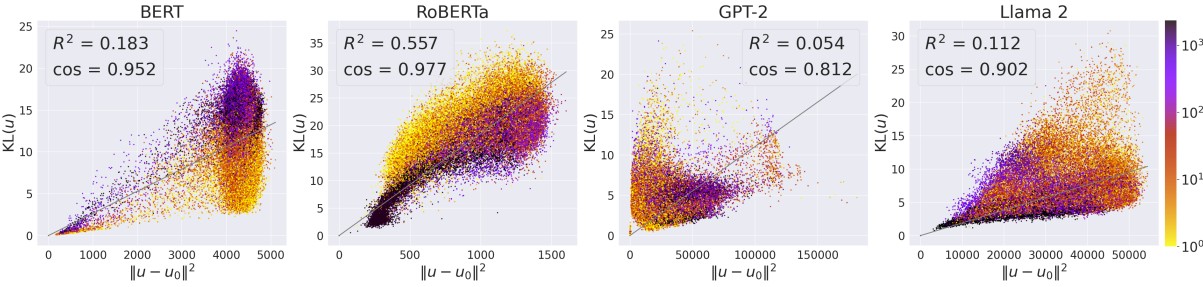

Figure 13: Linear relationship between the KL divergence and the squared norm of contextualized embedding for BERT, RoBERTa, GPT-2, and Llama 2. The color represents token frequency.

In the left panel, the linear relationship was not reproduced. Looking at the right panel, the norm of embedding reduces for low-frequency words with $n_w < 100$; plots of the same shape are also found in the literature (Schakel and Wilson, 2015; Arefyev et al., 2018; Pagliardini et al., 2018; Khodak et al., 2018). This is considered a consequence of insufficient optimization epochs; the norm of parameters tends to be smaller due to the implicit regularization (Arora et al., 2019), thus the trained parameters do not satisfy the ideal SGNS model (4) very well, particularly for low-frequency words.

## H Results on pre-trained word embeddings

In this section, we show that the linear relationship between the KL divergence and the squared norm of word embedding holds also for pre-trained word embeddings.

### H.1 Pre-trained fastText embeddings

We used Wiki word vectors provided by Bojanowski et al. (2017). These 300-dimensional embeddings are trained for 5 epochs on Wikipedia with the fastText model. We used the same KL divergence as in Appendix G, which was calculated on the Wikipedia dump corpus. Results are shown in the left panel of Figure 12, where we randomly selected 10,000 words that appeared not less than $10^4$ times in the Wikipedia dump.

### H.2 Pre-trained SGNS embeddings

We used pre-trained SGNS vectors provided by Li et al. (2017). These 500-dimensional embeddings are trained for 2 epochs on Wikipedia with the SGNS model. We used the same KL divergence as in Appendix G, which was calculated on the Wikipedia dump corpus. Results are shown in the right panel of Figure 12, where we randomly selected 10,000 words that appeared not less than $10^4$ times in the Wikipedia dump.

## I Results on contextualized embeddings

**Settings.** For the experiment of contextualized word embeddings, we used embeddings obtained from the final layer of BERT, RoBERTa, GPT-2, and Llama 2. We obtained 2000 sentences from One Billion Word Benchmark (Chelba et al., 2014) and input them into each language model to get contextualized embeddings of all tokens. Special

|  | raw | | whitened | |
|---|---|---|---|---|
|  | $R^2$ | cos | $R^2$ | cos |
| BERT | 0.183 | 0.952 | 0.003 | 0.898 |
| RoBERTa | 0.557 | 0.977 | 0.196 | 0.943 |
| GPT-2 | 0.054 | 0.812 | 0.431 | 0.905 |
| Llama 2 | 0.112 | 0.902 | 0.127 | 0.894 |

Table 10: Linear relationship strength between KL divergence and squared norm of language model contextual word embeddings. Presented are coefficients of determination ($R^2$) and uncentered correlation coefficients (cosine similarity) for both raw and whitened embeddings. Larger values indicate better performance.

tokens at the beginning and end of tokenized inputs, if any, were excluded.

**Results.** Looking at the scatterplots in Fig. 13, approximate linear relationships can be observed in BERT, RoBERTa, and Llama 2, but in GPT-2, the linear relationship is somewhat weaker. According to the values in Table 10, whitening improves the linear relationship for GPT-2 and Llama 2, but it worsens for BERT and RoBERTa, and the effect of whitening is not clear-cut. While there is still room for discussion, overall, an approximate linear relationship between KL divergence and the squared norm of contextual embeddings appears to hold.

## J Basic properties of the exponential family of distributions

**The expectation and covariance matrix.** The first and second derivatives of $\psi(u)$ are computed as

$$\frac{\partial \psi(u)}{\partial u} = e^{-\psi(u)} \frac{\partial}{\partial u} \sum_{w' \in V} q(w') e^{\langle u, v_{w'} \rangle}$$
$$= e^{-\psi(u)} \sum_{w' \in V} q(w') v_{w'} e^{\langle u, v_{w'} \rangle}$$
$$= \sum_{w' \in V} p(w'|u) v_{w'},$$

$$\frac{\partial^2 \psi(u)}{\partial u \partial u^\top} = \frac{\partial}{\partial u}\Big(e^{-\psi(u)} \sum_{w' \in V} q(w') v_{w'} e^{\langle u, v_{w'}\rangle}\Big)^\top$$

$$= e^{-\psi(u)} \frac{\partial}{\partial u} \sum_{w' \in V} q(w') v_{w'}^\top e^{\langle u, v_{w'}\rangle}$$

$$+ \frac{\partial e^{-\psi(u)}}{\partial u} \sum_{w' \in V} q(w') v_{w'}^\top e^{\langle u, v_{w'}\rangle}$$

$$= e^{-\psi(u)} \sum_{w' \in V} q(w') v_{w'} v_{w'}^\top e^{\langle u, v_{w'}\rangle}$$

$$- \frac{\psi(u)}{\partial u} e^{-\psi(u)} \sum_{w' \in V} q(w') v_{w'}^\top e^{\langle u, v_{w'}\rangle}$$

$$= \sum_{w' \in V} p(w'|u) v_{w'} v_{w'}^\top - \eta(u)\eta(u)^\top$$

$$= \sum_{w' \in V} p(w'|u)(v_{w'} - \eta(u))(v_{w'} - \eta(u))^\top,$$

showing (6) and (7), respectively.

**KL divergence.** For computing the KL divergence, first note that

$$\log \frac{p(w'|u_1)}{p(w'|u_2)}$$
$$= \langle u_1 - u_2, v_{w'}\rangle - \psi(u_1) + \psi(u_2)$$

from (4). Thus, the KL divergence is

$$\mathrm{KL}(p(\cdot|u_1) \,\|\, p(\cdot|u_2)) =$$
$$\sum_{w' \in V} p(w'|u_1)\Big(\langle u_1 - u_2, v_{w'}\rangle - \psi(u_1) + \psi(u_2)\Big)$$
$$= \langle u_1 - u_2, \eta(u_1)\rangle - \psi(u_1) + \psi(u_2), \qquad (18)$$

showing (8).

**Approximation of KL divergence.** Next, we consider the Taylor expansion of $\psi(u)$ at $u = u_1$. By ignoring higher order terms of $O(\|u - u_1\|^3)$, we have

$$\psi(u) \simeq \psi(u_1) + \frac{\partial \psi(u)}{\partial u^\top}\bigg|_{u_1} (u - u_1)$$
$$+ \frac{1}{2}(u - u_1)^\top \frac{\partial^2 \psi(u)}{\partial u \partial u^\top}\bigg|_{u_1} (u - u_1).$$

Using (6) and (7), we can rewrite this expression for $u = u_2$ as

$$\psi(u_2) \simeq \psi(u_1) + \langle u_2 - u_1, \eta(u_1)\rangle$$
$$+ \frac{1}{2}(u_2 - u_1)^\top G(u_1)(u_2 - u_1), \quad (19)$$

and substituting it into (18), we obtain

$$2\mathrm{KL}(p(\cdot|u_1) \,\|\, p(\cdot|u_2))$$
$$\simeq (u_1 - u_2)^\top G(u_1)(u_1 - u_2), \qquad (20)$$

showing (9) for $i = 1$. Considering the Taylor expansion of $G(u)$ at $u = u_2$, each element of $G(u_1)$ is $G_{ij}(u_1) = G_{ij}(u_2) + O(\|u_1 - u_2\|)$. Thus we can rewrite the right hand side of (20) as $(u_1 - u_2)^\top (G(u_2) + O(\|u_1 - u_2\|))(u_1 - u_2) \simeq (u_1 - u_2)^\top G(u_2)(u_1 - u_2) + O(\|u_1 - u_2\|^3)$. Therefore, we have shown that (9) holds for both $i = 1$ and $i = 2$.

## K  High-dimensional random vectors

**Random vector setting.** In this section, we adopt a probabilistic viewpoint and treat the elements of vectors $u$ and $v$ as random variables denoted by $u^i$ and $v^i$ for $i = 1, \ldots, d$ to estimate the orders of magnitude of various quantities, such as vector norms. Although the embedding vectors $\{u_w\}_{w \in V}$, $\{v_{w'}\}_{w' \in V}$ are not random variables, the random variable setting is justified when we randomly sample words $w$ and $w'$ from a large corpus and set $u = u_w$ and $v = v_{w'}$. To simplify the analysis, we assume that the vector elements are distributed independently. While we could relax this assumption by imposing the spherical condition (Jung and Marron, 2009; Aoshima et al., 2018), we leave this extension for future work.

We aim to discuss the relative magnitudes of vectors, so rescaling the vectors does not affect the argument. Therefore, we assume that each element is proportional to $d^{-1/2}$, i.e., $u^i = O_p(d^{-1/2})$, $v^i = O_p(d^{-1/2})$. The squared norm of $u$ is $\|u\|^2 = \sum_{i=1}^d (u^i)^2 = O_p(d \cdot (d^{-1/2})^2) = O_p(1)$, and the norm itself is also $\|u\| = (\|u\|^2)^{1/2} = O_p(1)$. Here $O_p(1)$ means that the magnitude of the vector remains bounded even if the dimension $d$ increases. The same applies to $v$, i.e., $\|v\| = O_p(1)$. The inner product of $u$ and $v$ is also $\langle u, v\rangle = \sum_{i=1}^d u^i v^i = O_p(d \cdot (d^{-1/2})^2) = O_p(1)$. Throughout this section, we consider magnitudes up to $O(d^{-1})$ and ignore higher order terms of $O(d^{-3/2})$ for sufficiently large $d$.

**Inner product with centered vector.** Each element of centered vector $u - \bar{u}$ is $u^i - \bar{u}^i = O_p(d^{-1/2})$, thus $\|u - \bar{u}\|^2 = \sum_{i=1}^d (u^i - \bar{u}^i)^2 = O_p(d \cdot (d^{-1/2})^2) = O_p(1)$. However, the inner product

$$\langle u - \bar{u}, v\rangle = O_p(d^{-1/2}), \qquad (21)$$

i.e., it tends to zero as $d \to \infty$. Similarly, $\langle u, v - \bar{v} \rangle = O_p(d^{-1/2})$. To show (21), note that $\mathbb{E}(u^i - \bar{u}^i) = \sum_{w \in V} p(w)(u_w^i - \bar{u}_w^i) = 0$. Thus, $\mathbb{E}((u^i - \bar{u}^i)v^i) = \mathbb{E}(u^i - \bar{u}^i)\mathbb{E}(v^i) = 0$. The variance is $\mathbb{E}(((u^i - \bar{u}^i)v^i)^2) = \mathbb{E}((u^i - \bar{u}^i)^2)\mathbb{E}((v^i)^2) = O(d^{-1} \cdot d^{-1}) = O(d^{-2})$. Therefore, $\mathbb{E}(\langle u - \bar{u}, v \rangle) = 0$, and $\mathbb{E}(\langle u - \bar{u}, v \rangle^2) = \mathbb{E}(\sum_{i=1}^d (u^i - \bar{u}^i)v^i)^2) = \sum_{i=1}^d \mathbb{E}(((u^i - \bar{u}^i)v^i)^2) + \sum_{i \neq j} \mathbb{E}((u^i - \bar{u}^i)v^i)\mathbb{E}((u^j - \bar{u}^j)v^j) = O(d \cdot d^{-2}) + 0 = O(d^{-1})$. This proves (21).

**$\bar{u}$ approximates $u_0$.** Regarding $v$, we used only the property $v^i = O_p(d^{-1/2})$ when deriving (21). So, the result does not change if we replace $v$ by $v - \bar{v}$: $\langle u - \bar{u}, v - \bar{v} \rangle = O_p(d^{-1/2})$. However, the result changes if we further replace $u$ by $u_0$:

$$\langle u_0 - \bar{u}, v - \bar{v} \rangle = O_p(d^{-1}), \qquad (22)$$

meaning that $\bar{u}$ approximates $u_0$. To show this, we first prepare another presentation of (5) as follows. Since $p(w') = q(w') \exp(\langle u_0, v_{w'} \rangle - \psi(u_0))$, (5) is expressed as $p(w'|u) = p(w') \exp(\langle u - u_0, v_{w'} \rangle - \psi(u) + \psi(u_0))$ by canceling out $q(w')$. We substitute $\psi(u)$ by (19) with $u_1 = u_0$, $u_2 = u$ to obtain

$$p(w'|u) \simeq p(w') \exp(\langle u - u_0, v_{w'} - \bar{v} \rangle$$
$$- \tfrac{1}{2}(u - u_0)^\top G(u - u_0)). \qquad (23)$$

In the above, $\langle u - u_0, v_{w'} - \bar{v} \rangle = O_p(d^{-1/2})$, and $(u - u_0)^\top G(u - u_0) = \sum_{w' \in V}(u - u_0)^\top (v_{w'} - \bar{v})(v_{w'} - \bar{v})^\top (u - u_0)p(w') = \sum_{w' \in V}\langle u - u_0, v_{w'} - \bar{v} \rangle^2 p(w') = O(d^{-1})$, because $\langle u - u_0, v_{w'} - \bar{v} \rangle = O_p(d^{-1/2})$.

Next, we consider (1) and let $p(w'|w) = p(w'|u_w)$ with (23).

$$p(w') = \sum_{w \in V} p(w'|u_w)p(w)$$
$$\simeq p(w') \sum_{w \in V} \exp\Big[\langle u_w - u_0, v_{w'} - \bar{v} \rangle$$
$$- \tfrac{1}{2}(u_w - u_0)^\top G(u_w - u_0)\Big]p(w).$$

This holds for any $w'$, thus $\sum_{w \in V} \exp[\cdots]p(w) \simeq 1$. By considering the Taylor expansion of the summand above, we have $\exp[\cdots] = 1 + \langle u_w - u_0, v_{w'} - \bar{v} \rangle - \tfrac{1}{2}(u_w - u_0)^\top G(u_w - u_0) + \tfrac{1}{2}\langle u_w - u_0, v_{w'} - \bar{v} \rangle^2 + O_p(d^{-3/2})$. Therefore, by taking

the summation, we have

$$\langle \bar{u} - u_0, v - \bar{v} \rangle$$
$$- \tfrac{1}{2} \sum_{w \in V}(u_w - u_0)^\top G(u_w - u_0)p(w)$$
$$+ \tfrac{1}{2} \sum_{w \in V}\langle u_w - u_0, v - \bar{v} \rangle^2 p(w) \simeq 0, \qquad (24)$$

where we have replaced $v_{w'}$ by $v$ to clarify that $w'$ is arbitrary. Here, $\sum_{w \in V}(u_w - u_0)^\top G(u_w - u_0)p(w) = O_p(d^{-1})$ and $\sum_{w \in V}\langle u_w - u_0, v - \bar{v} \rangle^2 p(w) = O_p(d^{-1})$, thus we have proved (22).

In addition to showing (22), we can also obtain an explicit formula for $\langle u_0 - \bar{u}, v - \bar{v} \rangle$. The second term in (24) is $\sum_{w \in V}(u_w - u_0)^\top G(u_w - u_0)p(w) = \sum_{w \in V} \text{tr} G(u_w - u_0)(u_w - u_0)^\top p(w) = \text{tr} GH$, where

$$H := \sum_{w \in V} p(w)(u_w - u_0)(u_w - u_0)^\top. \qquad (25)$$

The third term in (24) is $\sum_{w \in V}\langle u_w - u_0, v - \bar{v} \rangle^2 p(w) = \sum_{w \in V}(v - \bar{v})^\top (u_w - u_0)(u_w - u_0)^\top (v - \bar{v}) = (v - \bar{v})^\top H(v - \bar{v})$. Therefore, we obtain

$$\langle u_0 - \bar{u}, v - \bar{v} \rangle \simeq \tfrac{1}{2}(v - \bar{v})^\top H(v - \bar{v})$$
$$- \tfrac{1}{2}\text{tr} GH. \qquad (26)$$

Interstingly, (26) shows that all the context embeddings $\{v_{w'}\}_{w' \in V}$ are constrained to a qudractic surface in $\mathbb{R}^d$.

**Proof of** (12). First note that

$$(u - u_0)^\top G(u - u_0)$$
$$= (u - \bar{u} + \bar{u} - u_0)^\top G(u - \bar{u} + \bar{u} - u_0)$$
$$= (u - \bar{u})^\top G(u - \bar{u}) + (\bar{u} - u_0)^\top G(\bar{u} - u_0)$$
$$+ 2(u - \bar{u})^\top G(\bar{u} - u_0).$$

Using (22), the magnitude of the remaining terms is obtained as follows. $(\bar{u} - u_0)^\top G(\bar{u} - u_0) = \sum_{w' \in V}(\bar{u} - u_0)^\top (v_{w'} - \bar{v})(v_{w'} - \bar{v})^\top (\bar{u} - u_0)p(w') = \sum_{w' \in V}\langle \bar{u} - u_0, v_{w'} - \bar{v} \rangle^2 p(w') = O((d^{-1})^2) = O(d^{-2})$. Similarly, $(u - \bar{u})^\top G(\bar{u} - u_0) = \sum_{w' \in V}(u - \bar{u})^\top (v_{w'} - \bar{v})(v_{w'} - \bar{v})^\top (\bar{u} - u_0)p(w') = \sum_{w' \in V}\langle u - \bar{u}, v_{w'} - \bar{v} \rangle\langle v_{w'} - \bar{v}, \bar{u} - u_0 \rangle p(w') = O(d^{-1/2} \cdot d^{-1}) = O(d^{-3/2})$. Therefore, we have shown that

$$(u - u_0)^\top G(u - u_0)$$
$$= (u - \bar{u})^\top G(u - \bar{u}) + O_p(d^{-3/2}),$$

where the magnitude of $(u - u_0)^\top G(u - u_0)$ and $(u - \bar{u})^\top G(u - \bar{u})$ is $O_p(d^{-1})$, and $u$ is arbitrary $u_w$. Combining this with (11) proves (12).

## L  Technical details of the contextualized embeddings

We need only the following additional modifications. The equation (1) for the unigram distribution $p(w)$ is replaced by

$$p(\cdot) = \sum_{i=1}^{N} p(\cdot|u_i)/N.$$

The definition (25) for the matrix $H$ in Appendix K is replaced by

$$H := \sum_{i=1}^{N} (u_i - u_0)(u_i - u_0)^{\top}/N.$$

These modifications simply replace the average weighted by word frequency $p(w)$ on the vocabulary set $V$ with the simple average over $\{u_i\}_{i=1}^{N}$. For a sufficiently large corpus size $N$ of the training set, the distribution of $\{u_i\}_{i=1}^{N}$ is approximated by a density function $\pi(u)$ of contextualized embedding $u$. Therefore, the simple average is interpreted as the expectation with respect to $\pi(u)$. Consequently, we can also employ an alternate approach to the definition: $\bar{u} = \int u\pi(u)\,du$, $p(\cdot) = \int p(\cdot|u)\pi(u)\,du$ and $H = \int (u - u_0)(u - u_0)^{\top}\pi(u)\,du$.