# OpenReview forum: "Norm of Word Embedding Encodes Information Gain"
_EMNLP/2023/Conference — EMNLP 2023 Main_

### Official Review · Reviewer_Jf3f · 2023-08-04

**Soundness:** 4

**Excitement:**

3: Ambivalent: It has merits (e.g., it reports state-of-the-art results, the idea is nice), but there are key weaknesses (e.g., it describes incremental work), and it can significantly benefit from another round of revision. However, I won't object to accepting it if my co-reviewers champion it.

**Missing References:**

The following paper is strongly relative to this paper.

Keyword Extraction Using Word Co-occurrence.
Christian Wartena, Rogier Brussee, Wout Slakhorst.
2010 Workshops on Database and Expert Systems Applications.
https://ieeexplore.ieee.org/document/5592000


**Paper Topic And Main Contributions:**

This paper discusses a characteristic analysis of word embeddings. The main findings are as follows:
- (1) The squared norm of word embedding approximates the information gain.
- (2) The bias-corrected KL divergence and the norm of word embedding are suitable as a metric of word informativeness.
The experiments about three tasks, keyword extraction, proper noun discrimination, and hypernym discrimination, are conducted to verify the second point.



**Reasons To Accept:**

- This paper is well structured. The claims are clearly explained and easy to read.
- Discussing from both theoretical and empirical aspects.


**Reasons To Reject:**

- As mentioned in Limitations, it is not clear whether the findings in this paper hold in general.
- In the experiment about keyword extraction, TFIDF is not applied as a baseline measure.


**Reproducibility:**

4: Could mostly reproduce the results, but there may be some variation because of sample variance or minor variations in their interpretation of the protocol or method.

**Reviewer Confidence:**

3: Pretty sure, but there's a chance I missed something. Although I have a good feel for this area in general, I did not carefully check the paper's details, e.g., the math, experimental design, or novelty.

---

> ### Author Rebuttal · Authors · 2023-08-29
>
> Thank you for your comments and suggestions for improving the paper.
>
> ---
>
> # Question about Generality
>
> > As mentioned in Limitations, it is not clear whether the findings in this paper hold in general.
>
> Thank you for your question! The same properties hold true for general settings and language models.
>
> ## General Settings
>
> First, theoretically, the proof consists only of parameter-independent quantities such as inner products and conditional probabilities. The theoretical property holds regardless of dimensionality or word frequency distribution.
>
> Empirically, experiments have also confirmed the property that the norm of some word vectors learned with multiple parameters (different training corpora, different training algorithms, different dimensionality) encodes the KL divergence of a word (see l.373, Appendix H).
>
> Indeed, generality is a point of interest to many readers. In the camera ready version, I would like to add my reply to Section 4.2, Experiments on other embeddings.
>
> ## Language Models
>
> We have performed additional experiments to confirm that norm of BERT embedding also encodes information gain. Also, we discuss some theoretical implications.
>
> ### Additional Experiments
>
> #### Result:
> A linear relationship between the norm of BERT embedding and the KL divergence is confirmed. We report the correlation coefficient between the norm of BERT embedding and the KL divergence.
>
> |         | Layer 0 (static) | Layer 12 (dynamic) |
> | ------- | ---------------- | ------------------ |
> | BERT    | 0.59             | 0.63               |
>
> #### Settings
> - KL divergence: KL(word) was computed based on the co-occurrence information of the word in the Wikipedia dump. (The corpus is different, yes, but I used a substitute.) The words are space-separated.
>
> - BERT embeddings: The BERT embeddings for a word are computed based on the respective tokenizer. For example, if the word is tokenized as wo, ##rd, then the average of the two embeddings is used as the word embedding. The embeddings of a word change depending on the context, so we finally averaged them to compute an embedding for a word.
>
> ### Theoretical Implications
>
> Since the language model and SGNS are compatible, we expect that the theory in this paper roughly corresponds to the language model. In the final layer of the language model, the probability of each word is calculated from the embedding u in the form of softmax(W u), so each row of W corresponds to v_{w'} for each word w' in SGNS. Since the probability of a word w' is proportional to exp( + bias_{w'}), the same form can be obtained by writing bias}_{w'} = log(nu q(w')) in the SGNS equation (4). Therefore, the SGNS theory in this paper is expected to be valid for language models as well.
> There are technical differences in the learning process, as described below, and we would like to discuss the details again.
>
> - (Difference 1) In the language model, $u$ is conditioned on word sequences before $w'$, but in SGNS, it is conditioned on only one surrounding word.
> - (Difference 2) The language model uses maximum likelihood and reinforcement learning, but SGNS uses NCE.
> - (Difference 3) In the language model, $\mathrm{bias}_{w'}$ is a tunable parameter, but in SGNS, $\nu$ and $q(w')$ are constants.
> - (Difference 4) The language model is softmax, but SGNS learns with the restriction that the denominator of softmax is a constant value.
>
> Certainly the interpretability of language models is an important topic and subject to interests of a wider readers. We thank you for stimulating this discussion! In the camera ready version, we would like to emphasize the result that our analysis method can be applied to language models.
>
> ---
>
> # Question about Experimental Settings of Keyword Extraction
>
> > In the experiment about keyword extraction, TFIDF is not applied as a baseline measure.
>
> Thank you for pointing that out. TFIDF is indeed a basic method for keyword extraction, but we did not use it as a baseline this time because we experimented in a special setting to investigate the specific properties of the KL divergence. We would like to provide an additional explanation for this.
>
> In order to show that the KL divergence includes the information necessary for keyword extraction, we conducted the experiment using the setting: "information on a specific document only, without referring to the document set". Therefore, the TFIDF that refers to the document set was not used as a baseline.
>
> In the camera ready version, we would like to reflect our reply at the beginning of Section 6.1.
>
> ---
>
> # Missing Reference
>
> > The following paper is strongly relative to this paper. Keyword Extraction Using Word Co-occurrence. Christian Wartena, Rogier Brussee, Wout Slakhorst. 2010 Workshops on Database and Expert Systems Applications. https://ieeexplore.ieee.org/document/5592000
>
> Thank you for pointing this out. The literature you mentioned is indeed very relevant to the metrics of word informativeness in this paper, since this study is based on the idea that words with special co-occurrences are considered informative words. In the camera ready version, we cite it in Section 2.1.

---

### Official Review · Reviewer_w5CX · 2023-08-11

**Soundness:** 4

**Excitement:**

3: Ambivalent: It has merits (e.g., it reports state-of-the-art results, the idea is nice), but there are key weaknesses (e.g., it describes incremental work), and it can significantly benefit from another round of revision. However, I won't object to accepting it if my co-reviewers champion it.

**Paper Topic And Main Contributions:**

In this paper, the authors find that the squared norm of SGNS word embeddings encodes the information gain of a word, which is defined by the KL divergence of the conditional probability distribution to the marginal distribution. The authors explain their findings theoretically through properties of the exponential family distribution, and verify their findings experimentally.

**Questions For The Authors:**

a. Does the norm of other word embeddings, such as GloVe or the word embedding layer in various deep learning models, also encode information gain?

b. Can the theoretical findings in the paper guide pratical work such as model design or model evaluation in the field of NLP?

**Reasons To Accept:**

- Previous works start from an empirical point of view, and believes that the norm of word embeddings represents the importance of words. In this paper, the authors theoretically analyze the relationship between the norm of word embeddings and the KL divergence of the
conditional probability distribution of the corresponding words in the context to the marginal distribution of the context. This idea is novel.

- The authors prove their argument from theory and experiments, providing sufficient support for their claim.

- Experiment results in the paper could be easily reproduced.

**Reasons To Reject:**

- As stated at the end of the paper, all theoretical analysis and experiments in the paper are limited to SGNS word embeddings. However, the SGNS method has been proposed for 10 years. In addition, the static word embeddings like SGNS and GloVe is not currently the mainstream method of NLP. So the overall contribution maybe relatively limited.

- The authors conducts theoretical analysis and experimental verification of the properties of SGNS word embeddings. Due to space limitations, it is understandable to only conduct theoretical analysis on SGNS. However, whether word embeddings obtained by other methods have similar properties should be verified in the experiments.


**Reproducibility:**

5: Could easily reproduce the results.

**Reviewer Confidence:**

4: Quite sure. I tried to check the important points carefully. It's unlikely, though conceivable, that I missed something that should affect my ratings.

---

> ### Author Rebuttal · Authors · 2023-08-29
>
> Thank you for your comments and suggestions for improving the paper.
>
> ---
>
> # Questions about Other Word Embeddings and Deep Learning Models
>
> > Does the norm of other word embeddings, such as GloVe or the word embedding layer in various deep learning models, also encode information gain?
>
> Thank you for your question. Our answer is separated in two parts.
>
> ## Other Word Embeddings
>
> We understood your question as whether the norm can be interpreted in the same way for word embeddings obtained by other methods. It could not be shown in the GloVe case, but it could be shown in fastText. Experiments have also confirmed the property that the norm of some word vectors learned with fastText encodes the KL divergence of a word (see l.373, Appendix H).
>
> ## Deep Learning Models
>
> We understood your question as whether the norm can be interpreted in the same way for embeddings in the language model. Our answer is yes. We have performed additional experiments using BERT to confirm this. In addition, we would like to discuss some theoretical implications.
>
> ### Additional Experiments
>
> #### Result:
>
> A linear relationship between the norm of BERT embedding and the KL divergence is confirmed. We report the correlation coefficient between the norm of BERT embedding and the KL divergence.
>
> |         | Layer 0 (static) | Layer 12 (dynamic) |
> | ------- | ---------------- | ------------------ |
> | BERT    | 0.59             | 0.63               |
>
> #### Settings
> - KL divergence: KL(word) was computed based on the co-occurrence information of the word in the Wikipedia dump. (The corpus is different, yes, but I used a substitute.) The words are space-separated.
>
> - BERT embeddings: The BERT embeddings for a word are computed based on the respective tokenizer. For example, if the word is tokenized as wo, ##rd, then the average of the two embeddings is used as the word embedding. The embeddings of a word change depending on the context, so we finally averaged them to compute an embedding for a word.
>
> ### Theoretical Implications
>
> Since the language model and SGNS are compatible, we expect that the theory in this paper roughly corresponds to the language model. In the final layer of the language model, the probability of each word is calculated from the embedding u in the form of softmax(W u), so each row of W corresponds to v_{w'} for each word w' in SGNS. Since the probability of a word w' is proportional to exp( + bias_{w'}), the same form can be obtained by writing bias}_{w'} = log(nu q(w')) in the SGNS equation (4). Therefore, the SGNS theory in this paper is expected to be valid for language models as well.
> There are technical differences in the learning process, as described below, and we would like to discuss the details again.
>
> - (Difference 1) In the language model, $u$ is conditioned on word sequences before $w'$, but in SGNS, it is conditioned on only one surrounding word.
> - (Difference 2) The language model uses maximum likelihood and reinforcement learning, but SGNS uses NCE.
> - (Difference 3) In the language model, $\mathrm{bias}_{w'}$ is a tunable parameter, but in SGNS, $\nu$ and $q(w')$ are constants.
> - (Difference 4) The language model is softmax, but SGNS learns with the restriction that the denominator of softmax is a constant value.
>
> Certainly the interpretability of language models is an important topic and subject to interests of a wider readers. We thank you for stimulating this discussion! In the camera ready version, we would like to emphasize the result that our analysis method can be applied to language models.
>
> ---
>
> # Question regarding the Practical Applicability
>
> > Can the theoretical findings in the paper guide pratical work such as model design or model evaluation in the field of NLP?
>
> Thank you for your question. We think that the norm of word embeddings provides information about the word informativeness, which could be used when interpreting the behavior of the model. How exactly to use it is a subject for future research.

---

### Official Review · Reviewer_Lew5 · 2023-08-11

**Soundness:** 4

**Excitement:**

4: Strong: This paper deepens the understanding of some phenomenon or lowers the barriers to an existing research direction.

**Paper Topic And Main Contributions:**

The authors investigate the word2vec skip-gram with negative sampling method on the type of information that is encoded.
They find that the squared norm of a vector approximates KL divergence. Further, they show that the squared norm of a word vector and KL divergence with bias correction work similarly well as a measure for the information a word carries. They theoretically corroborate their findings and back them further up with experiments.
Using their findings, they show that Kullback-Leibler divergence as well as the squared norm of word vectors provide appropriate measures for the information a word carries wrt to specific tasks.

**Questions For The Authors:**

- In the derivation, (9) ignores the higher order terms with the condition that u_1 is close to u_2. Similarly, (10) considers u_w is close to u_0 with u_0 set to be stopwords? Then, u_0 is replaced by u-head for practical computation. Can we justify that u_w is close to u-head, or even u_0?
- The derivation shows that KL(w) of a word correlates to the square of the norm of the word embedding with the whitening-like transformation. However, this transformation is not used in the following experiments. The authors should consider adding an explanation on the choice.
- Is it possible to apply this method for the explainability of (larger) language models? Can we trace the "strength" of informativeness of the single embedded tokens through the layers of a network?

**Reasons To Accept:**

- The paper is set up well and presents a clear story, setting the findings of the authors into context.
- The authors consider the word-frequency bias of KL divergence and work with a bias-correction version of KL and embedding norm. The bias-corrected squared norm of word embedding shows great improvement compared to its baselines, especially on the proper-noun discrimination tasks.

**Reasons To Reject:**

- It is not clear if the finding still holds for different parameters choices of training the embeddings.
- Since most work nowadays focuses on "contextualized embeddings", this would be maybe more interesting to a bigger community -- however, using contextualized versions of words might make it more complicated.

**Reproducibility:**

4: Could mostly reproduce the results, but there may be some variation because of sample variance or minor variations in their interpretation of the protocol or method.

**Reviewer Confidence:**

2: Willing to defend my evaluation, but it is fairly likely that I missed some details, didn't understand some central points, or can't be sure about the novelty of the work.

**Typos Grammar Style And Presentation Improvements:**

- The paper is very well written and follows a clear structure.

Typos:

- ll. 477: ".. explained in Appendix B."

---

> ### Author Rebuttal · Authors · 2023-08-29
>
> Thank you for your comments and suggestions for improving the paper.
>
> ---
>
> # Question about Generality
>
> > It is not clear if the finding still holds for different parameters choices of training the embeddings.
>
> Thank you for your suggestion.
>
> When training word vectors, one can choose different parameters. I understood your question to be whether our findings are general findings that are robust to such changes. The answer is yes.
>
> First, theoretically, the proof consists only of parameter-independent quantities such as inner products and conditional probabilities. The theoretical property holds regardless of dimensionality or word frequency distribution.
>
> Empirically, experiments have also confirmed the property that the norm of some word vectors learned with multiple parameters (different training corpora, different training algorithms, different dimensionality) encodes the KL divergence of a word (see l.373, Appendix H).
>
> Indeed, generality is a point of interest to many readers. In the camera ready version, I would like to add my reply to Section 4.2, Experiments on other embeddings.
>
> ---
>
> # Questions regarding the Contextualized Embeddings and Language Models
>
> > Since most work nowadays focuses on "contextualized embeddings", this would be maybe more interesting to a bigger community -- however, using contextualized versions of words might make it more complicated.
>
> > Is it possible to apply this method for the explainability of (larger) language models? Can we trace the "strength" of informativeness of the single embedded tokens through the layers of a network?
>
> Thank you for your question. We understood your question about whether the norm can be interpreted the same way for embeddings in the language model. Our answer is YES. We have performed additional experiments to confirm this. In addition, we would like to discuss some theoretical implications.
>
> ## Additional Experiments
> ### Result:
> A linear relationship between the norm of BERT embedding and the KL divergence is confirmed. We report the correlation coefficient between the norm of BERT embedding and the KL divergence.
>
> |  | Layer 0 (static) | Layer 12 (dynamic) |
> |----------|----------|----------|
> | BERT | 0.59 | 0.63 |
>
>
> ### Settings:
>
> - KL divergence: KL(word) was computed based on the co-occurrence information of the word in the Wikipedia dump. (The corpus is different, yes, but I used a substitute.) The words are space-separated.
> - BERT embeddings: The BERT embeddings for a word are computed based on the respective tokenizer. For example, if the word is tokenized as wo, ##rd, then the average of the two embeddings is used as the word embedding. The embeddings of a word change depending on the context, so we finally averaged them to compute an embedding for a word.
>
> ## Theoretical Implications
>
> Since the language model and SGNS are compatible, we expect that the theory in this paper roughly corresponds to the language model. In the final layer of the language model, the probability of each word is calculated from the embedding u in the form of softmax(W u), so each row of W corresponds to v_{w'} for each word w' in SGNS. Since the probability of a word w' is proportional to exp(<v_{w'}, u> + bias_{w'}), the same form can be obtained by writing bias_{w'} = log(nu q(w')) in the SGNS equation (4). Therefore, the SGNS theory in this paper is expected to be valid for language models as well.
>
> There are technical differences in the learning process, as described below, and we would like to discuss the details again.
>
> - (Difference 1) In the language model, $u$ is conditioned on word sequences before $w'$, but in SGNS, it is conditioned on a surrounding word.
> - (Difference 2) The language model uses the maximum likelihood method and reinforcement learning, but SGNS uses NCE.
> - (Difference 3) In the language model, bias_{w'} is a tunable parameter, but in SGNS, $\nu$ and $q(w')$ are constants.
> - (Difference 4) The language model is softmax, but SGNS is trained with the restriction that the denominator of softmax is a constant value.
>
> Certainly the interpretability of language models is an important topic and subject to interests of a wider readers. We thank you for stimulating this discussion! In the camera ready version, we would like to emphasize the result that our analysis method can be applied to language models.
>
> ---
>
> # Question about Theoretical Analysis
>
> > In the derivation, (9) ignores the higher order terms with the condition that u_1 is close to u_2. Similarly, (10) considers u_w is close to u_0 with u_0 set to be stopwords? Then, u_0 is replaced by u-head for practical computation. Can we justify that u_w is close to u-head, or even u_0?
>
> Thank you for pointing out the unclear points in the paper. In the original paper,  "close" was used in two different meanings, which was misleading, so I will explain the difference clearly when revising the manuscript.
>
> In equation (9), u1 and u2 are "not very far from each other" due to the approximation that ignores higher-order terms. On the other hand, in equation (10), the stop words u_w and u_0 are "very close" and we assume that the stop words u_w = u_0 with very small errors (which is also experimentally verified). What was difficult to understand in the original manuscript was that equation (11), like equation (9), intended that u_w and u_0 for the general words w were "not very far from each other". In the camera ready version, this explanation will be added to clear up this last point.
>
> ---
>
> # Question regarding the Whitening-like transformation
>
> > The derivation shows that KL(w) of a word correlates to the square of the norm of the word embedding with the whitening-like transformation. However, this transformation is not used in the following experiments. The authors should consider adding an explanation on the choice.
>
> Thank you very much for your suggestion. It would certainly be easier to understand if it is explained why the whitening-like transformation used in the theoretical analysis was not used in the following experiments. The explanation is as follows.
>
> In the experiment, we used embedding without any transformation to ensure that the word informativeness is encoded in the norm of the word vector that is practically used. The experimental results are comparable for both.
>
> ---
>
> # Typos
>
> > ll. 477: ".. explained in Appendix B."
>
> Thank you for pointing this out, we will fix it in the camera ready version.

---

### Meta-Review · Area_Chair_P7TJ · 2023-09-15

**Recommendation:** 5

**Metareview:**

This paper demonstrates a relationship between the norm of word embeddings, the extent to which a word's co-occurrence probabilities diverge from the unigram distribution ("information gain"), and the informativeness of the word with respect to several tasks.

The reviewers agreed that the research was sound. Opinions were generally positive on "excitement." One concern, which I shared, was that the results pertain only to static word embeddings. During the discussion period, the authors added a follow-up experiment showing similar results for contextualized embeddings (presumably averaged?).

---

### Decision · Program_Chairs · 2023-10-07

**Decision:**

Accept-Main

**Comment:**

This paper demonstrates a relationship between the norm of word embeddings, the extent to which a word's co-occurrence probabilities diverge from the unigram distribution ("information gain"), and the informativeness of the word with respect to several tasks.

The reviewers agreed that the research was sound. Opinions were generally positive on "excitement." One concern, which I shared, was that the results pertain only to static word embeddings. During the discussion period, the authors added a follow-up experiment showing similar results for contextualized embeddings (presumably averaged?).